



# Isothermal evaporation of α-pinene secondary organic aerosol particles formed under low-NOx and high-NOx conditions

Zijun Li[1], Angela Buchholz[1], Luis M.F. Barreira[1,2], Arttu Ylisirniö[1], Liqing Hao[1], Iida Pullinen[1], Siegfried Schobesberger[1], Annele Virtanen[1, *]

[1]Department of Applied Physics, University of Eastern Finland, Kuopio, Finland

[2]Atmospheric Composition Research, Finnish Meteorological Institute, Helsinki, Finland

*Correspondence to*: Annele Virtanen (annele.virtanen@uef.fi)

**Abstract.** Many recent secondary organic aerosol (SOA) studies mainly focus on biogenic SOA particles formed under low-NOx conditions and thus are applicable to pristine environments with minor anthropogenic influence. Although interactions between biogenic volatile organic compounds and NOx are important in, for instance, suburban areas, there is still a lack of knowledge about volatility and processes controlling the evaporation of biogenic SOA particles formed in the presence of high concentrations of NOx. Here we provide detailed insights into the isothermal evaporation of α-pinene SOA particles that were formed under low-NOx and high-NOx conditions to investigate the evaporation process and the evolution of particle composition during the evaporation in more detail. We coupled Filter Inlet for Gases and AEROsols-Chemical Ionization Mass Spectrometer (FIGAERO-CIMS) measurements of the molecular composition and volatility of the particle phase with isothermal evaporation experiments conducted under a range of relative humidity (RH) conditions from dry to 80 % RH. Very similar changes were observed in particle volatility at any set RH during isothermal evaporation for the α-pinene SOA particles formed under low-NOx and high-NOx conditions. However, there were distinct differences in the initial composition of the two SOA types, possibly due to the influence of NOx on the RO2 chemistry during SOA formation. Such compositional differences consequently impacted the primary type of aqueous-phase processes in each type of SOA particles in the presence of particulate water.

## 1 Introduction

Biogenic secondary organic aerosol (SOA) affects the climate by scattering solar radiation (Lihavainen et al., 2015) and acting as cloud condensation nucleic (Sporre et al., 2014; Yli-Juuti et al., 2021). It overall causes a negative radiative forcing on climate and potentially offsets approximately 13 % of the warming effect due to a doubling of $CO_2$ (Sporre et al., 2019). Nonetheless, due to its short atmospheric lifetime (i.e., days to weeks) (Hodzic et al., 2016), biogenic SOA cannot be simply regarded as a long-term offset to the warming impact from the long-lived $CO_2$ (i.e., lifetime of years) (Myhre et al., 2013). Accurate assessments of the radiative forcing from biogenic SOA require a comprehensive understanding about the key processes affecting its atmospheric lifetime. As one of those processes controlling aerosol lifetime, evaporation governs the gas-particle partitioning of organic compounds. The recent findings of evaporation studies are primarily applicable to pristine forest areas (Vaden et al., 2011; Yli-Juuti et al., 2017; Buchholz et al., 2019; Li et al., 2019; Li et al., 2021), hence there is still a lack of information about the evaporation behavior of biogenic SOA particles in polluted areas where anthropogenic–biogenic interactions are important. Here, we investigate the isothermal evaporation of biogenic SOA particles which are derived from oxidizing one representative biogenic compound (BVOC) precursor in the presence of nitrogen oxides ($NO_x \equiv NO+NO_2$) that are important gaseous pollutants. The results of this study will help redefine the atmospheric lifetime of biogenic



SOA in, for instance, suburban areas that have experienced rapid growth of $CO_2$ emissions in the recent decade (Mitchell et al., 2018).

BVOCs account for up to 90 % of the global budget of non-methane volatile organic compounds (Guenther et al., 1995). Atmospheric oxidation of BVOCs produces condensable vapors and subsequently contributes to the formation of biogenic

SOA (Ehn et al., 2014; Jokinen et al., 2015). With annual emissions of approximately 157–177 Tg, monoterpenes ($C_{10}H_{16}$) have been considered as an important class of BVOCs (Guenther et al., 2012) and recognized as a significant source of biogenic SOA not only in boreal forest areas (Tunved et al., 2006; Mohr et al., 2019; Barreira et al., 2021) but also in certain isoprene-dominated regions (Zhang et al., 2018; Lee et al., 2020). As monoterpene with the highest emissions from terrestrial vegetation, α-pinene has been commonly used as a proxy for monoterpenes. Oxidizing α-pinene with ozone ($O_3$) or hydroxyl radicals (OH)

produces SOA particles with high particle viscosities (Virtanen et al., 2010; Saukko et al., 2012) and substantial amounts of oxidation products of low volatility (Lopez-Hilfiker et al., 2015; Ylisirniö et al., 2020). Compared to semi-volatile organic mixtures, α-pinene SOA particles display slower evaporation rates, which are attributed to the existence of compounds of low volatility, high particle viscosity and possible particle-phase reactions (Vaden et al., 2011; Yli-Juuti et al., 2017; D'Ambro et al., 2018). While kinetic limitations arising from particle viscosity hinder particle evaporation especially under dry conditions,

the volatility distributions of particulate constituents primarily control the particle evaporation rate at high relative humidity (RH). On the one hand, particulate water content acts a plasticizer accelerating the bulk-surface diffusion (Yli-Juuti et al., 2017; Li et al., 2019). On the other hand, it behaves as a catalyst to induce aqueous-phase processes which form organic compounds of low volatility especially for SOA particles with high oxidization levels (Buchholz et al., 2019; Li et al., 2021). All these findings primarily advance our knowledge about the evaporation processes of biogenic SOA particles but limited to those in

remote forest areas lacking in anthropogenic influence.

Over the past few decades, increasing amounts of $NO_x$ emissions have been released into the atmosphere due to rapid economic growth and urbanization (Duncan et al., 2016). In recent years, increasing attention has been paid to the chemical interactions between $NO_x$ and BVOC emissions, especially monoterpenes in suburban areas. In the southern United States, increasing amounts of daytime $NO_x$ were suggested to enhance the formation of monoterpene SOA and increase the likelihood of

fragmentation chemistry that forms oxidation products with carbon number ($C_{num}$) smaller than 10 (Zhang et al., 2018). In the same location, the large nighttime production of monoterpene SOA was attributed to oxidation induced by $NO_3$ which is a product of $NO_2$ and $O_3$ (Xu et al., 2015). In the southwest Germany, $NO_x$ accumulated at night drove the large production of multifunctional organic nitrates via the nocturnal chemistry between monoterpenes and $NO_x$, contributing up to 25 % in mass to the particulate organics (Huang et al., 2019). Furthermore, hydrolysis which converts organic nitrates into $HNO_3$ has been

suggested as an efficient sink for organic nitrates in the particle phase (Zare et al., 2018; Zare et al., 2019). Depending on the alkyl substitutions, functional groups, and carbon backbones, the hydrolysis mechanism and rates can vary between different monoterpene-derived organic nitrates (Wang et al., 2021). Overall, the addition of $NO_x$ affects the formation of monoterpene SOA, the types of oxidation products, and potentially the properties of the resulted particles. For monoterpene SOA particles produced in the presence of $NO_x$, there is very limited knowledge about the evolution of particle properties during isothermal

evaporation process.

In this study, we investigated the isothermal evaporation of SOA particles from oxidizing α-pinene under low-$NO_x$ and high-$NO_x$ conditions at room temperature with a wide range of RH. We examined the volume change as well as the evolution of the molecular composition of α-pinene SOA particles during isothermal evaporation. While most recent laboratory studies focusing on the evaporation process of biogenic SOA particles represent conditions relevant for pristine environments with

minor anthropogenic influence, our work would be one of the few providing insights into the SOA evaporation under suburban settings involving anthropogenic-biogenic interactions.


## 2 Methods

### 2.1 Isothermal evaporation of SOA particles

An oxidation flow tube reactor (OFR) of 13.3 L was used to generate α-pinene SOA particles at ~ 40 RH % and 25 °C with a
residence time of 160 seconds (Kang et al., 2007; Lambe et al., 2011). Experimental conditions and results of SOA generation
are summarized in Table S1. The operation procedure of this OFR has been described previously (Buchholz et al., 2019; Li et
al., 2021). Briefly, α-pinene was continuously injected into a heated flow of $N_2$ with a syringe pump (Kari et al., 2018) and
then mixed with a humidified flow of $N_2$, $O_2$, $O_3$, and $N_2O$ before entering into the OFR. Before being mixed with $O_3$, the
mixing ratio of α-pinene was constantly monitored by a proton transfer reaction time-of-flight mass spectrometer (PTR-TOF
8000, Ionicon Analytik) using hydronium ($H_3O^+$) ions. Overall, 5 L min$^{-1}$ of total flow containing α-pinene (250 – 300 ppb)
and $O_3$ (~ 13 ppm) was introduced into the OFR for photooxidation. Inside the OFR, SOA was formed by oxidizing α-pinene
by OH radicals, which were produced by photolyzing $O_3$ (~ 13 ppm) with 254-nm UV lamps in the presence of water vapor.
In the high-NO$_x$ case, NO and $NO_2$ were produced in situ via the photolysis of $N_2O$ (99.5 % purity, mixing ratio inside OFR:
1.85 % in volume) with the same 254-nm lamps inside the OFR (Lambe et al., 2017). The OH exposure was estimated to be
$(2.6 \pm 0.3) \times 10^{11}$ and $(1.72 \pm 0.07) \times 10^{11}$ molec cm$^{-3}$ for low-NO$_x$ and high-NO$_x$ conditions after considering the external OH
reactivity (Peng et al., 2015; Peng et al., 2016). For the high-NO$_x$ case, the estimated ratio between the [RO$_2$] + [NO] and
[RO$_2$] + [HO$_2$] pathways ($\frac{[RO_2]+[NO]}{[RO_2]+[HO_2]}$) was $1.30 \pm 0.18$. For each experiment under one NO$_x$ condition, similar aerosol mass
concentration in the OFR was ensured. Assuming a particle density of 1.5 g cm$^{-3}$, the mass concentrations of polydisperse α-
pinene SOA were $442 \pm 30$ and $139 \pm 29$ μg cm$^{-3}$ under low-NO$_x$ and high-NO$_x$ conditions, respectively. Between experiments,
the OFR was flushed with purified air overnight with the same RH and illumination as the following experiment but without
adding α-pinene and $N_2O$. Afterwards, the background particle number and the VOC mixing ratio were reduced to be below
2000 # cm$^{-3}$ (mass concertation < 0.1 μg cm$^{-3}$) and under the instrument limit of detection, respectively. Despite the careful
cleaning procedure, we cannot exclude a small contamination of the system with $N_2O$ (e.g., in tubing before the OFR) and/or
HONO (a by-product of the in-situ NO$_x$ production from $N_2O$) which may lead to a small production of NO$_x$ in experiment
where no $N_2O$ was added ([NO$_x$] < 8 ppb at the outlet of OFR). Hence, we labeled the experiments with no $N_2O$ addition as
"low-NO$_x$" and expect a very small contribution of NO$_x$-related compounds.

After the OFR, 2 L min$^{-1}$ of the generated α-pinene SOA was passed through an ozone denuder coated with potassium iodide,
and then size selected with two parallel nanometer aerosol differential mobility analyzers (NanoDMA, model 3085, TSI)
operated in an open-loop setting with a sample-to-sheath flow ratio of 1:8 or 1:10. During the size selection, the gas vapors
were diluted by at least 2 orders of magnitude so that particle evaporation was initiated (Li and Chen, 2005). By varying the
ratio of dry and humidified flows for the sheath flow of the NanoDMAs, the RH of particle samples was set to one of three
desired conditions: dry (< 7 %), intermediate RH (40 % RH), or high RH (80 % RH). Afterwards, the size-selected SOA
particles with 80-nm electrical mobility diameters were fed into either bypass lines with varying length or stainless-steel
residence time chambers (RTC) with volumes of 25 L or 100 L for isothermal evaporation, allowing residence times ($t_R$) from
seconds to hours. The vapor wall losses in the RTC were sufficiently fast with a vapor wall loss coefficient greater than $10^{-2}$ s$^{-1}$, ensuring negligibly low concentrations of gas vapors in the RTC (Yli-Juuti et al., 2017). After each isothermal evaporation
experiment, the NanoDMAs, bypass lines and RTCs were flushed for at least 12 h with purified air under the same RH
condition as the following experiment.





### 2.2 Physical and chemical characterization of SOA particles

**2.2.1 SMPS analysis**

Particle size changes due to isothermal evaporation were periodically measured with a scanning mobility particle sizer (SMPS, TSI classifier model 3080, CPC 3775). We quantified the extent of particle isothermal evaporation using the volume fraction remaining (VFR) under the assumption of particle sphericity. Independent of changes in particle number or mass concentration, the VFR was determined as $(D_{p,t}/D_{p,0})^3$, where $D_{p,t}$ and $D_{p,0}$ are the measured size after residence time $t$ of isothermal

evaporation and the selected sizes at the start of isothermal evaporation, respectively. The selected size $D_{p,0}$ was calibrated against ammonium sulfate particles under the experimental RH conditions.

**2.2.2 HR-ToF-AMS analysis**

Elemental composition measurements of particle samples were conducted using a high-resolution time-of-flight aerosol spectrometer (HR-ToF-AMS, Aerodyne Research Inc.). The oxidation state ($OS_c = 2$ O:C – H:C) of particle samples was

calculated with the improved ambient parameterizations (Canagaratna et al., 2015). Following the Eqs. (1) – (2) presented in Kiendler-Scharr et al. (2016), we estimated the mass concentration of organic nitrate moieties on the basis of the ratio of $NO_2^+$ and $NO^+$. The values of $NO_2^+/NO^+$ for ammonium nitrate (i.e., from the ionization-efficiency calibration) and organic nitrates were correspondingly set to 0.51 and 0.1. We also estimated the upper bound of organic nitrates by assuming all observed nitrates are in the organic form. When sampling from the RTC during evaporation, particle mass concentration was not

sufficient to derive reliable elemental composition values from the AMS measurements. Thus, we only presented the elemental composition data from particle samples right after size selection which experienced the least amount of particle evaporation.

**2.2.3 FIGAERO-CIMS analysis**

Composition and volatility of particle samples were characterized with a custom-built Filter Inlet for Gases and AEROsols (FIGAERO) (Ylisirniö et al., 2021) in combination with a high-resolution time-of-flight chemical ionization mass spectrometer

(CIMS, Aerodyne Research Inc.) using the iodide ionization scheme (Lopez-Hilfiker et al., 2014). The operation of FIGAERO-CIMS has been detailed in previous studies (Lopez-Hilfiker et al., 2014; Ylisirniö et al., 2021). The FIGAERO inlet allows sequential gas and particle measurements using two individual inlets leading to the CIMS. Since we were interested in the changes in particle composition and volatility, the system was only deployed for measuring particle-phase compounds. The mass resolution of CIMS was between 4000 and 5000 and the ion molecule reaction chamber was actively controlled at a

pressure of 100 mbar. Due to the lack of a reliable sensitivity and transmission calibration for the relevant compounds detected with FIGAERO-CIMS, ion signals are presented in counts per second (ct s$^{-1}$), i.e., variabilities in sensitivity and transmission between compounds were not accounted for.

During sample collection, particles were sampled either after size selection (i.e., fresh stages with avg. $t_R = 0.25$ h) or after isothermal evaporation in the RTC (i.e., RTC stages with avg. $t_R = 4.25$ h) onto a PTFE filter (2 μm pore, Zefluor, Pall Corp.)

for 0.5 h. The collected particles were thermally desorbed from the filter into the CIMS using a temperature-controlled dry N$_2$ flow. The temperature of the N$_2$ flow was linearly increased from 25 °C to ~ 200 °C within 20 min (i.e., desorption period) and then kept at ~200 °C for an extra 15 min (i.e., soak period) so that organic residuals, if any, could evaporate from the filter. Apart from particle samples, we also performed two different types of blank measurements to identify instrument background and potential contaminations from the sampling process. The instrument background was investigated by measuring clean

FIGAERO filters with no particle collection, while additional contaminations were determined by analyzing samples which were collected through the NanoDMAs with a set voltage of 0 V (i.e., no selected particles) and 0.5-h collection time.





For each observed ion, the change in signal intensity with desorption temperature ($T_{desorp}$) is called thermogram. The desorption temperature with the maximum signal ($T_{max}$) is proportional to the saturation vapor concentration ($C^*$, defined at 25 °C) of a compound (Lopez-Hilfiker et al., 2014). In this study, the $T_{max} – C^*$ relationship was calibrated using a homologous series of

polyethylene glycol compounds (PEG 4 - 8) as reference compounds (Ylisirniö et al., 2021). According to the volatility classification defined by Donahue et al. (2012), the range of $T_{desorp}$ was divided into semi-volatile organic compounds (SVOCs), low-volatility organic compounds (LVOCs) and extremely low volatility organic compounds (ELVOCs). Note that the conducted calibration is sufficient to reliably identify the SVOC and LVOC ranges, despite uncertainties in extrapolating the calibration towards lower $C^*$ values (Hyttinen et al., 2022). The measured calibration parameters are listed in the Supplement.

**2.2.4 Positive matrix factorization analysis**

Positive matrix factorization (PMF) (Paatero and Tapper, 1994) was deployed to analyze the mass spectra data collected by the FIGAERO-CIMS (Buchholz et al., 2020; Li et al., 2021). Organic compounds which have similar temporal behavior during isothermal evaporation were grouped into a single factor. Blank samples were also included in the data sets for PMF analysis to determine background factors. Error matrices were prepared using the constant error scheme (Buchholz et al., 2020) without

any down-weighting, and the PMF results were evaluated with the PMF Evaluation Tool (PET v3.05) (Ulbrich et al., 2009). Due to the predominance of non-nitrated organics in all particle samples (Table 1), the PMF analysis was independently applied to non-nitrated organics and organic nitrates so that the organic nitrates could be better resolved into factors. This approach also allowed us to investigate potential similarities between the low-NO$_x$ and high-NO$_x$ experiments regarding the non-nitrated organic compounds. After careful comparisons between multiple PMF solutions, we decided to select PMF solutions with 12

and 8 factors for non-nitrated organics and organic nitrates data sets, respectively. We assigned each PMF factor as a sample or background factor according to its contribution in the particle and filter blank samples. For the non-nitrated organics data set, 5 sample and 7 background factors were found. For the organic nitrate data set, 3 out of the 8 factors were identified as sample factors. In the following discussion, only the sample factors will be considered, but the factor thermograms and mass spectra of all factors are shown in the Supplement. Note that the seemingly large number of background factors is due to

changes in the exact composition of the instrument background between different experiment days (e.g., due to changing the PTFE filter in the FIGAERO inlet), i.e., each sample has contributions from only 2 or 3 out of all background factors. Detailed information on the PMF solution diagnostics can be found in the Supplement.

**3 Results and Discussion**

**3. 1 Isothermal evaporation and thermal desorption behavior of SOA particles**

Changes in the VFR of particles as a function of $t_R$ at different RH are presented for low-NO$_x$ (grey) and high-NO$_x$ (orange) conditions in Figure 1. The particle evaporation rates were almost identical between the two NO$_x$ conditions at any set RH. We acknowledge that certain amounts of inorganic nitrates (e.g., NH$_4$NO$_3$, HNO$_3$) were formed under high-NO$_x$ conditions due to the high levels of gaseous HNO$_3$ and HONO which were byproducts of the NO production via N$_2$O photolysis (Lambe et al., 2017). However, estimated from the AMS data, the produced inorganic nitrates contributed approximately 1 % to the

particle mass (Table 1) and would only lead to very minor changes in particle volume during isothermal evaporation (i.e., smaller than the measurement uncertainty). With increasing RH, the particle evaporation rate became faster (Figure 1), and a larger fraction of compounds was removed during isothermal evaporation. As previous particle evaporation studies have suggested (Vaden et al., 2011; Yli-Juuti et al., 2017; Buchholz et al., 2019; Li et al., 2019; Zaveri et al., 2020; Li et al., 2021), considerable kinetic limitations exist in dry SOA particles and thus evaporation of compounds is slowed down due to the

substantially high particle viscosity. With increasing RH, particulate water acts as a plasticizer for the bulk – surface diffusion and therefore enhances particle evaporation. In addition, the almost identical particle evaporation rates between intermediate-



RH (40 % RH, diamonds) and high-RH (80 % RH, circles) conditions suggest that the particle evaporation at 40 % RH can be already approximated as a liquid-like process (Figure 1). For the investigated SOA particles, we observed that particulate water also acted as a catalyzer for aqueous-phase processes (see section 3.4). Note that decoupling the plasticization and catalytic roles of particulate water on the SOA particle evaporation requires the development of process-level models considering particle-phase chemistry and is thus beyond the scope of this work.

In Figure 2, we show the thermal desorption profiles for the high-$NO_x$ case at fresh and RTC evaporation stages under dry (Figure 2c, d) and high-RH (Figure 2a, b) conditions. Thermal desorption behavior of particle samples is usually illustrated with sum thermograms (STGs) in which the total sum of organic signals is plotted against $T_{desorp}$ (Lopez-Hilfiker et al., 2015; D'Ambro et al., 2017). Note that the appearances of STGs are impacted by both the total number of molecules collected onto the filters and the underlying volatility distributions. To allow feasible comparisons between fresh and RTC samples at the same RH, all STGs were at first normalized to the total ion signals of each individual sample. Following the same procedure described by Li et al. (2021), we then scaled the normalized STGs of the RTC samples with the changes in the average VFR ($VFR_{avg}$) between fresh and RTC stages to account for the expected particle volume change due to isothermal evaporation. To enhance readability, we will hereafter call both – the normalized STGs for the fresh samples and the scaled normalized STGs of the RTC samples – "STG", since they are now directly comparable.

The difference between STGs of fresh and RTC samples (Figure 2a, b; indicated as gray-striped areas) became larger when particulate water was present. Overall, organic nitrates accounted for up to 20 % of the total organic signals of FIGAERO samples under high-$NO_x$ conditions, which broadly agrees with the values estimated from the AMS data under the same condition (Table 1). Since thermal losses of nitrate moieties from organic nitrates during FIGAERO desorption are possible (Francisco and Krylowski, 2005), the estimated fraction of organic nitrates here represents the lower bound. We also examine changes in the median desorption temperature values ($T_{50}$) for the total organics, non-nitrated organics, and organic nitrates between different samples, as shown in Figure 2e. On average, the organic nitrates were slightly more volatile and had lower values of $T_{50}$, as compared to the non-nitrated organics. In line with the faster particle evaporation rate at high RH (Figure 1) and larger removal of volatile compounds (Figure 2a-d), changes in $T_{50}$ were more noticeable in the high-RH samples than the dry ones for all compositional categories (Figure 2e). Very similar changes in STGs and $T_{50}$ were observed in the low-$NO_x$ case in the Supplement.

### 3.2 PMF of SOA particles

In the following sessions, we presented the independent PMF analysis results for the non-nitrated organics and organic nitrates. The integrated mass spectra showing the overall composition for each particle sample are presented in the Supplement.

### 3.2.1 Characterization of non-nitrated organics

As shown in Figure 3, five sample factors (i.e., F1 – 5) were identified for non-nitrated organics. Ratios between sample factors changed when $NO_x$ was present during SOA formation in the OFR. Compared with non-nitrated organic factors in the high-$NO_x$ samples, the organic nitrates (in purple; sum of the sample factors of the organic nitrates data set, see section 3.2.2) overall spanned a volatility range similar to F2 and F3. Note that small amounts of organic nitrates were observed in the low-$NO_x$ sample with a signal contribution of 2 % or less to the particle sample. This was probably caused by small contaminations of the OFR system with $N_2O$ from the preceding high-$NO_x$ experiments. All non-nitrated organic factors except F4 mainly consisted of compounds with a molecular weight (MW) of less than 250 Da and with $C_{num}$ of 10 or less (Figure 4). For the factors F1 – 4, the factor volatility which is indicated by the $T_{50}$ values (Table 2) exhibited stronger dependence on the signal-weighted MW rather than the signal-weighted $OS_c$. Moreover, contributions of compounds with $C_{num}$ of 10 or more to the factor spectra become more important with increased values of $T_{50}$. Since the increase of oxygen and hydrogen atoms



counterbalances the simultaneous extension of carbon backbones, $OS_c$ does not reflect the effect of MW on the factor volatility in this case. Even though F5 is the least volatile sample factor, the range of MW and $C_{num}$ of the compounds grouped into F5 was very similar to that of F2. The combination of high $T_{50}$ and the dominance of compounds with $C_{num}$ of 10 or less in F5

very likely suggests that these compounds are largely the products from the thermal decomposition of (extremely) low-volatility parent compounds during the FIGAERO desorption.

### 3.2.2 Characterization of organic nitrates

We identified three sample factors (i.e., NF1 – 3) from organic nitrates, as shown in Figure 5. While NF1 – 3 only occurred in the high-$NO_x$ samples, an additional sample factor was found in the low-$NO_x$ ones. The formation of this unique sample factor

in the low-$NO_x$ samples was possibly due to the very small amount of $NO_x$ being formed in the low-$NO_x$ experiments. Since the total contribution of the organic nitrates in the low-$NO_x$ samples was no more than 2 %, we here only focus on the analysis of the high-$NO_x$ samples of which up to 20 % of particle mass were attributed to organic nitrates (Table 1). Accounting for the nitrogen content of organic nitrates, we calculated the $OS_c$ (i.e., $OS_c = 2 \cdot \frac{O}{C} - \frac{H}{C} - 5 \cdot \frac{N}{C}$) of each nitrogen-containing compound under the assumption that all nitrogen existed in the form of alkyl nitrates (Priestley et al., 2021). While NF2 and

3 appeared under all conditions, the contribution of NF1 to particle samples became significant only in the high-RH samples (Figure 5a), suggesting that the formation pathway of NF1 was different from the other two organic nitrate factors'. All the mass spectra of NF1 – 3 were dominated by compounds with $C_{num}$ of 10 or less, although noticeable amounts of compounds with $C_{num} > 10$ were observed in NF3 (Figure 5b), consistent with its overall high value of $T_{desorp}$ suggesting the lowest volatility among the organic nitrate factors.

### 3.3 Effect of $NO_x$ on the α-pinene SOA formation

The signal contributions of non-nitrated organic sample factors to the total sum of organics were compared between low-$NO_x$ (in grey) and high-$NO_x$ cases (in orange) in Figure 6a. Here, only the dry particle samples collected at the fresh stages were chosen for analysis, as they were subject to the least amount of isothermal evaporation during the FIGAERO sample collection. Substantially large differences in the signal contributions were observed for multiple sample factors. Under low-$NO_x$ condition,

F1, F2, and F5 were the three major factors. For the high-$NO_x$ samples, F3 was the dominant factor followed by approximately equal contributions from the other factors. Small but noticeable differences in the factor contributions were observed for F4 between the two $NO_x$ conditions. The observed difference in the factor mass contribution between the investigated particle samples could be induced by the enhancement/suppression of formation pathways for certain compounds (Kroll and Seinfeld, 2008) and/or the gas–particle equilibrium partitioning which follows Raoult's law (Donahue et al., 2006). According to the

partitioning theory (Pankow, 1994) for a system in equilibrium between gas and particle phase, the $C^*$ of a sample factor $k$ can be expressed as

$$C_k^* = OA \cdot \frac{G_k}{P_k}, \tag{1}$$

where OA is the organic particle mass concentration, which was determined from SMPS data, by assuming a particle density of 1.5 μg m$^{-3}$. $G_k$ and $P_k$ represent the mass concentrations of a sample factor $k$ in gas and particle phase, respectively.

Rearranging Eq. (1), we can calculate the total mass concentration of a sample factor $k$ in both gas and particle phase as follows

$$G_k + P_k = (OA + C_k^*) \cdot \frac{P_k}{OA} \tag{2}$$

The $C^*$ value of a sample factor $k$ can be estimated by converting its $T_{50}$ value into $C^*$ at 25 °C with a parametrization derived from the $T_{max} - C^*$ calibration. $\frac{P_k}{OA}$ is the mass contribution of a sample factor $k$ in the initial particles prior to isothermal


evaporation. Here, we use the mass fraction of a sample factor $k$ in the dry, fresh samples as the proxy of $\frac{P_k}{OA}$, since these particle

samples experienced the minimum amount of isothermal evaporation.

The total mass concentration in gas and particle phase for each sample factor $(G_k + P_k)$, derived using Eq. (2), is shown in Figure 6b. Addition of $NO_x$ during SOA formation inside the OFR decreased the production (i.e., the sum of gas and particle mass concentrations) of the compounds grouped into F1, F2, and F5, and increased the production of organic nitrates. Therefore, changes in their individual contributions to the high-$NO_x$ sample were primarily caused by the effect of $NO_x$ on the formation

of the compounds in these factor groups. Furthermore, it should be noted that for F3 and F4, their individual productions were comparable between the low-$NO_x$ and high-$NO_x$ cases. Thus, in comparison with the low-$NO_x$ sample, the enhanced contributions of F3 and F4 to the high-$NO_x$ sample were most likely due to the decreased contributions of other factors in the particle phase.

Note that the total mass concentration estimated in Eq. (2) was under the assumption of the gas – particle equilibrium during

the timescale of the OFR. Since particles were always produced at 40 % RH where negligible kinetic limitation exists, equilibrium should be reached between gas and particle phase during the minutes of residence in the OFR (Li and Shiraiwa, 2019). Another caveat to the discussion above is the potential occurrence of particle-phase reactions during SOA formation inside the OFR. Many of these reactions can be acid-catalyzed. Note that $HNO_3$ is formed as a byproduct when using $N_2O$ to generate $NO_x$ in the OFR (Lambe et al., 2017). Ranney and Ziemann (2016) reported that gaseous $HNO_3$ (0.25 – 2.0 ppm) was

able to catalyze the particle-phase hemiacetal dehydration for alkane oxidation products. However, this type of reaction would not be relevant with respect to the OFR residence time, due to the negligible amount of $HNO_3$ produced in the OFR and its very long reaction timescale of weeks. For non-catalyzed reactions, their reaction rates depend on their own reaction rate constants as well as both gas vapor concentration and condensation rate (Peng and Jimenez, 2020). Since information on the kinetics of non-catalyzed reactions is still rare, their importance on the SOA formation in the OFR is difficult to estimate.

**3.4 Evolution of PMF factors during isothermal evaporation**

By coupling the sample factor contributions from the FIGAERO measurement with changes in $VFR_{avg}$ retrieved from the SMPS measurements during isothermal evaporation, we are able to derive the net change ratio (NCR) to evaluate the net transformation (i.e., material loss vs. production) of individual sample factors during isothermal evaporation (Li et al., 2021). Each factor in each particle sample is compared to its reference case (i.e., its contribution in the dry, fresh condition). If a low-

volatility factor simply remains in the particle phase (i.e., no change) or the formation and removal processes are equal to each other (i.e., no net change), NCR is equal to 1. NCR values > 1 mean that the contribution of the factor increases due to an actual formation process. NCR values < 1 indicate that the contribution of the factor decreases either due to evaporation or because the compounds grouped into this factor were consumed in particle-phase reactions. To determine if evaporation is the likely cause for the change, we use the characteristic desorption temperature (characteristic $T_{desorp}$) of the factor which is the

desorption temperature at which 25 %, 50 %, and 75 % of the factor thermogram signal is reached. This measure gives a deeper insight into the shape of the thermogram than just using $T_{50}$ and is a better representation of the volatility range of a factor.

**3.4.1 NCR of non-nitrated organics**

The changes in NCR were minor in the dry samples for most sample factors (Figure 7b), which is consistent with the small change in VFR (Figure 1), STGs (Figure 2a-d), and particle composition (Figure 3),. Only F1, the sample factor with the lowest

characteristic $T_{desorp}$ (Figure 7a), displayed a different trend between the low-$NO_x$ and high-$NO_x$ cases, with a decrease in the NCR in the low-$NO_x$ sample but almost no change in the high-$NO_x$ one (Figure 7b). The NCR of organic nitrates (i.e., sum of NF1 – 3) decreased with isothermal evaporation under dry conditions. Loss due to evaporation seems to be unlikely as they





spanned a volatility range from LVOCs to ELVOCs. It is possible that when organic nitrates were able to decompose into HNO$_3$ and non-nitrated organic products during isothermal evaporation, the remaining organics might still be retained in the dry particle samples due to high particle viscosity and possibly contributed to F1 in the high-NO$_x$ sample.

When particulate water was present, i.e., under high-RH conditions, the evolution of NCR became more complex. F1 already showed a lower value of NCR at the fresh stage compared to the dry conditions, and it was no longer present in the particle after isothermal evaporation in the RTC (indicated by cross symbols in Figure 7b). As the volatility of F1 was high enough to allow significant evaporation within 4.25 h, its decreasing NCR can still be primarily caused by evaporation. Similar to the case of F1, F2 and F4 also showed lower values of NCR in the presence of water as compared to the dry conditions. Contrary to the dry conditions, a clear decreasing trend in NCR of these factors with decreasing VFR$_{avg}$ was observed under high-RH conditions already at the fresh stage. Since little evaporation is expected for compounds in the volatility range of LVOCs and ELVOCs to which F2 and F4 were assigned (Li et al., 2019), there must be another mechanism (i.e., aqueous-phase processes) primarily driving the loss of these compounds at the fresh stage with a timescale in the order of minutes. With increasing evaporation time at high RH, the NCR for each factor diverged between SOA types. In the high-NO$_x$ case, the NCR values of F2 and F4 further decreased with evaporation time, which suggests that the material loss mechanism continued as the dominant pathway controlling their evolution during isothermal evaporation. In the low-NO$_x$ case, however, their NCR values showed an increasing trend during isothermal evaporation, indicating that the chemical production of the grouped compounds outweighed the loss pathway at this stage. Although the NCR trends of F2 were different between the low-NO$_x$ and high-NO$_x$ cases under high-RH conditions, the characteristic $T_{desorp}$ increased in both cases (Figure 7a). Such an increase was interpreted as an indication for aqueous-phase chemistry in a previous study (Buchholz et al., 2020). This may indicate, that for both cases, the presence of particulate water was crucial, but the extent or speed of the transformation varied.

When investigating F3 at high RH, the change in its NCR was dependent on the NO$_x$ condition during SOA formation. Already for the fresh stage at high RH, the NCR of F3 showed a pronounced increase and became much larger than 1 in the low-NO$_x$ case. However, the NCR of F3 stayed more or less at 1 in the high-NO$_x$ case. As F3 falls in the range of LVOCs and ELVOCs (similar to F2), only little evaporation is expected within the time span of that case (avg. t$_R$ = 0.25 h) (Li et al., 2019). Consequently, there must be an efficient production pathway under the high-RH condition that contributes to compounds assigned to F3 already at the fresh stage. The corresponding thermograms (red in Fig. 3, 3$^{rd}$ row) are alike in the low-NO$_x$ and high-NO$_x$ cases, suggesting that this production had occurred in both samples, but was more dependent on particulate water in the low-NO$_x$ sample than in the high-NO$_x$ one. A small but noticeable decrease due to isothermal evaporation is expected for compounds within the range of LVOCs for the timescale of 4.5 h (Li et al., 2019). When the low-NO$_x$ particles continued to evaporate at high RH, the NCR of F3 indeed decreased. Considering the change in its factor thermogram shape (Figure 3a) and the minor shift in its characteristic $T_{desorp}$ (Figure 7a) in the low-NO$_x$ case at high RH, it is very likely that the decrease in the NCR of F3 during isothermal evaporation was mostly driven by the evaporation of its volatile content with high $C^*$. In the high-NO$_x$ samples, on the other hand, the NCR of F3 barely changed with evaporation time, even under high RH. In this case, there must be a production pathway, which compensated for the material loss for F3 in the high-NO$_x$ sample.

Unlike the dry conditions, F5 already showed noticeable changes in its NCR at the fresh stage under high-RH condition. Since little evaporation was expected for compounds in the ELVOC range in the time span of hours (Li et al., 2019), the observed changes in the NCR of F5 in either NO$_x$ case would suggest the existence of aqueous-phase processes already at the fresh stage. Similar to F3, the evolution of F5 with increasing evaporation also varied between the two particle types at high RH. With a decreasing VFR$_{avg}$ (i.e., increasing isothermal evaporation), F5 showed a decrease in its NCR in the low-NO$_x$ samples but exhibited an increase in its NCR in the high-NO$_x$ ones. Such different evolution trends of the NCR of F5 between the two NO$_x$ levels at high RH indicate that the competition between production and loss reactions in the aqueous phase varied with the SOA formation conditions.





**3.4.2 Possible aqueous-phase processes affecting NCR of non-nitrated organics**

When compounds dissociate (here, mostly hydrolyze) into smaller molecules, the products of this process can either evaporate from or stay in the particle phase and possibly participate in further reactions with other molecules. Note that products that evaporate do not contribute anymore to any of the factors. Linking two or more molecules via chemical bonds (accretion reactions) produces larger compounds of lower volatility which are less likely to evaporate. These products can undergo further

reactions with increasing isothermal evaporation time especially in the presence of water. Note that if they have sufficiently low volatility, newly formed compounds from aqueous-phase processes will remain in the particle phase and contribute to other factors.

In the high-$NO_x$ samples, only decreasing or constant NCR values were observed for factors F2 – F5 at the fresh stage under high-RH conditions. This suggests that the reactions consuming F2 and F4 are mostly hydrolysis type processes (i.e., creating

products that do not remain in the particle phase). The formation pathways for products grouping into F3 and F5 are minor. This formation of more volatile compounds may increase the observed isothermal evaporation of the particles. In the low-$NO_x$ case, we observed factors with increasing NCR (F3 and F5) and with decreasing NCR (F2 and F4) when RH increased. It is likely that the particle-phase reaction involving compounds in F2 and/or F4 result in low volatility products contributed to F3 and F5. Hence, accretion type reactions must be the dominant pathway for this SOA type. Products from such accretion

reactions could have sufficiently low volatility and slow down the particle evaporation, possibly resulting in the slightly larger $VFR_{avg}$ in the low-$NO_x$ case (Figure 7c). One possible type of accretion reaction could be the Baeyer – Villiger reaction of peroxides (e.g., peroxycarboxylic acid and hydroperoxide) (Lim and Turpin, 2015; Claflin et al., 2018) which are expected to be relatively more abundant for the low-$NO_x$ samples of which $RO_2 + HO_2$ reactions were favored during SOA formation, compared to the high-$NO_x$ samples (Ziemann and Atkinson, 2012; Peng and Jimenez, 2020). Note that $RO_2 + HO_2$ reactions

can also lead to the production of carboxylic acids. Although we cannot rule out the occurrence of Fischer esterification requiring carboxylic acids, such reaction could not be observed even in extremely acidic particles for the relevant time scales of hours (Birdsall et al., 2013; Kristensen et al., 2014).

When particles continued to evaporate in the RTC under high-RH conditions, there were again differences in the evolution of F2 – F5 for low-$NO_x$ and high-$NO_x$ conditions. While F2 and F4 displayed continuous decreases in NCR in the high-$NO_x$

samples, these two factors experienced increases in their NCR in the low-$NO_x$ ones. Very likely, hydrolysis and accretion were the underlying reactions for the high-$NO_x$ case, and those resulted products of sufficient low volatility possibly contributed to the constant and increasing NCR values of F3 and F5. For the high-$NO_x$ samples, the important role of accretion at the RTC stage but not at the fresh stage might indicate the timescale required for such reaction to be in the order of hours. Kinetic measurements have suggested that while the peroxy hemiacetal formation involving peroxides and carbonyls has a timescale

from 1 min to 2h , the formation of hemiacetal from reactions between alcohol and carbonyls is at least one order of magnitude slower (i.e., timescale of 10h) (Ziemann and Atkinson, 2012). Furthermore, the increasing NCR of F2 and F4 in the low-$NO_x$ case was driven by the net production of compounds. This was likely attributed to the decomposition of compounds with short lifetime (e.g., acylperoxy hemiacetals and peroxy hemiacetals), originally (i.e., at the fresh stage) grouped into F3 and F5 (Claflin et al., 2018).

**3.4.3 NCR of organic nitrates**

Already under dry conditions, decreasing values of NCR were observed for NF2 and NF3 after isothermal evaporation in the RTC. As some compounds grouped into NF2 desorbed in the SVOC range (Figure 5a ,1[st] and 2[nd] rows), loss due to evaporation is a possible explanation for the behavior of NF2. However, the volatility of NF3 was in the ELVOC range and so little evaporation is expected even under high RH (Li et al., 2019). Thus, the decreasing NCR for NF3 under dry conditions can



possibly be attributed to the chemical decomposition of the organic nitrates. When particles were conditioned to RH below 10 % in the dry experiments, it is possible that a small amount of water was retained in the particle phase (Ghorai et al., 2011; Lin et al., 2021) and available for catalyzing the decomposition of organic nitrates. Due to the diffusion limitations caused by high particle viscosity under dry conditions, some of the remaining organics after the decomposition of NF3 might stay in the samples and be grouped into any of the non-nitrated organic factors. As the non-nitrated organic factor F1 was in the volatility range between SVOCs and LVOCs, moderate evaporation could still take place in the timescale of 4.25 h even when the particle viscosity is high (Li et al., 2019). Therefore, the negligible change in the NCR of F1 (Figure 7b, red) cannot be explained without considering the additional formation of compounds (i.e., products from the decomposition of NF2 and/or NF3).

When large amounts of particulate water were present, the NCR values of NF1 – 3 distinctly changed with increasing evaporation. While NF1 displayed a significant increase in its NCR at the fresh stage under high-RH conditions, NF2 and NF3 showed notable decreases in their NCR. Since both NF2 and NF3 belonged to the LVOC and ELVOC ranges, very minor evaporation would be expected for them within the average timescale of 0.25 h at the fresh stage (Li et al., 2019). Thus, the pronounced reductions in their NCR were mostly driven by aqueous-phase processes. Even though the NCR of NF2 and NF3 also showed a decreasing trend with increasing evaporation under dry conditions, we did not observe any production of compounds grouped into NF1 (Figure 5a) or an increase in its NCR (Figure 8b). One explanation for the difference in the NCR of NF1 could be that the functional groups involved in the decomposition process vary between the two evaporation stages (dry, RTC vs. high RH, fresh). As the produced organic nitrates are multifunctional, hydrolysis is not necessarily restricted to the nitrate groups under high-RH conditions. Other functional groups like peroxides and esters can undergo hydrolysis as well. On the one hand, compounds in NF1 could be products from the decomposition of NF2 and NF3. On the other hand, they could be also formed via the nitration of alcohols with $HNO_3$ that could be produced from the decomposition of NF2 and NF3 (Wang et al., 2021). When SOA particles continued to evaporate in the timescale of 4.25 h at high RH, compounds within the range of SVOCs could be possibly removed by evaporation due to their high $C^*$ values. However, significant removal primarily due to evaporation would not be expected for those compounds in the range of LVOCs and ELVOCs (Li et al., 2019), unless other loss mechanisms exist. As a majority of compounds in each organic nitrate factor were desorbed in this volatility range, their decreasing NCR values were highly likely caused by another loss process (i.e., aqueous-phase process) in addition to evaporation.

## 4 Conclusions

Many studies have focused on the impact of $NO_x$ on the aerosol yield as well as gaseous and particulate products during SOA formation (Zhao et al., 2018; Pullinen et al., 2020). Our study is one of the few studies providing molecular-level information about particle-phase processes in SOA that were formed in the presence of $NO_x$. We examined the changes in volume and molecular composition of α-pinene SOA particles that were formed under low-$NO_x$ and high-$NO_x$ conditions during isothermal evaporation.

Although up to 20 wt % of the particle-phase material could be attributed to organic nitrates in the high-$NO_x$ case, the overall particle volatility derived from isothermal evaporation and from FIGAERO-CIMS measurements was very similar for the two SOA types. Applying PMF to analyze the FIGAERO-CIMS thermal desorption data revealed distinct differences in the initial composition of both the particle and gas phase of the two SOA types for the non-nitrated organic compounds. These observations may be explained by that the effect of high $NO_x$ concentrations during the oxidation of α-pinene goes beyond the formation of organic nitrates. High $NO_x$ concentrations could also suppress the formation of organic compounds (e.g., peroxide-containing compounds) from the $RO_2 + HO_2$ reaction pathways. Apart from competing with $RO_2 + HO_2$ reactions,



recent work suggests that with increased $NO_x$ emissions, the $RO_2 + NO$ pathway can also surpass autoxidation reactions during the photooxidation of α-pinene and thus decrease the yield of highly oxygenated molecules (HOMs) (Pye et al., 2019). The majority of them partition into the particle phase due to their low volatilities (Mutzel et al., 2015). This change in HOM formation will both decrease the aerosol yield and change the composition of the formed SOA. HOMs often contain multiple peroxide functional groups (Bianchi et al., 2019). Thus, the reduction of HOM production will further lower the overall

peroxide content of SOA particles together with the suppression of $RO_2 + HO_2$ reactions under high $NO_x$ conditions.

At higher RH, both SOA types exhibited faster particle evaporation rates and lost a larger fraction of materials as compared to dry conditions. This agrees with previous studies showing that higher RH enhances particle evaporation (Yli-Juuti et al., 2017; Buchholz et al., 2019; Li et al., 2019). But in our study, we find that although the enhancement in isothermal evaporation was similar for both SOA types, the dominant aqueous-phase processes occurring in the particle phase were indeed different. I.e.,

the differences in the initial particle composition (e.g., the lower peroxide content for the high $NO_x$ conditions) led to different chemical reactions occurring in the particle phase when particulate water was present. Many of the compounds involved in the aqueous-phase processes were of low or extremely low volatility, and hence did not evaporate in the RTC within the average experimental time scale of 4.25 h. This suggests that the observed process-level differences are potentially important for the fate of individual organic molecules in the particle phase, but their impact on overall particle volatility under ambient

conditions might be negligible.

Interactions of BVOC emissions with high concentrations of $NO_x$ are especially relevant in suburban areas. In such environments, the particle composition and aqueous-phase processes may be different from those found in low-$NO_x$ environments, for instance in pristine forest areas where particulate samples should have higher peroxide contents (Surratt et al., 2006). As our findings are limited to the aqueous-phase process in α-pinene SOA particles, future studies on other BVOC

systems and even emissions from urban vegetation are needed to estimate the overall importance of these processes and how they may affect other physicochemical properties of SOA particles.

**Data availability.** The data set is available upon request from Annele Virtanen (annele.virtanen@uef.fi).

**Supplement.** The supplement related to this article is available online.

**Author contribution.** ZL, AB, and AV designed the study. ZL, AB, LB, AY, and LH carried out laboratory experiments. ZL,

AB, LH, IP, SS, and AV performed data analysis and interpretation. ZL wrote the paper with contributions from all coauthors.

**Competing interests.** The authors declare that they have no conflict of interest.

**Acknowledgements.** This research has been supported by the Academy of Finland (grant nos. 299544, 310682, 307331, and 317373), the Itä-Suomen Yliopisto (Doctoral Programme in Environmental Physics, Health and Biology), and FP7 Ideas: European Research Council (QAPPA, grant no. 335478).




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





**Table 1**. Mass contribution of non-nitrated organics and organic nitrates to α-pinene SOA particle samples under high-NO$_x$ conditions.

| Particle Sample | Instrument | Method | Non-nitrated organics (wt %) | Organic nitrates (wt %) |
|---|---|---|---|---|
| Dry Fresh | FIGAERO-CIMS | | 82 | 18 |
| | AMS [a] | NO$_2^+$/NO$^+$ [b] | 92 | 7 |
| | | ON$_{max}$ [c] | 89 | 11 |
| Dry RTC | FIGAERO-CIMS | | 86 | 14 |
| | AMS | | N/A | N/A |
| Intermediate RH Fresh | FIGAERO-CIMS | | N/A | N/A |
| | AMS | NO$_2^+$/NO$^+$ | 90 | 9 |
| | | ON$_{max}$ | 88 | 12 |
| High RH Fresh | FIGAERO-CIMS | | 82 | 18 |
| | AMS | NO$_2^+$/NO$^+$ | 79 | 19 |
| | | ON$_{max}$ | 77 | 23 |
| High RH RTC | FIGAERO-CIMS | | 93 | 7 |
| | AMS | | N/A | N/A |

[a] For AMS methods, the average molecular weight of organic nitrates estimated from the FIGERO – CIMS data was used.

[b] NO$_2^+$/NO$^+$: Estimating organic nitrate contribution using the difference in fragmentation between organic and inorganic nitrate species (Kiendler-Scharr et al., 2016).

[c] ON$_{max}$: Assuming that all nitrate moieties detected in the AMS were from organic nitrates.

**Table 2**. Summary of non-nitrated organic factors in α-pinene SOA particle samples under low-NO$_x$ and high-NO$_x$ conditions

| ID | Composition | OS$_c$ | MW / g mol$^{-1}$ | $T_{50}$ and interquartile range of $T_{desorp}$ [a] | |
|---|---|---|---|---|---|
| | | | | Low-NO$_x$ | High-NO$_x$ |
| F1 | $C_{6.9}H_{9.8}O_{4.9}$ | 0.21 | 171.05 | 76.07 [65.70, 92.53] | 69.03 [60.93, 78.08] |
| F2 | $C_{7.7}H_{10.1}O_{5.6}$ | 0.25 | 192.15 | 103.15 [88.42, 122.45] | 95.42 [82.71, 110.30] |
| F3 | $C_{8.7}H_{12.8}O_{5.4}$ | -0.13 | 203.67 | 103.68 [89.24, 130.50] | 99.78 [82.03, 128.14] |
| F4 | $C_{10.9}H_{14.5}O_{7.1}$ | 0.07 | 258.98 | 136.70 [119.99, 156.37] | 138.37 [119.96, 159.14] |
| F5 | $C_8H_{10.8}O_{5.4}$ | 0.18 | 193.26 | 146.57 [123.89, 168.13] | 157.43 [140.78, 173.65] |

[a] Values of $T_{50}$ and interquartile range of $T_{desorp}$ were determined with the dry, fresh samples.






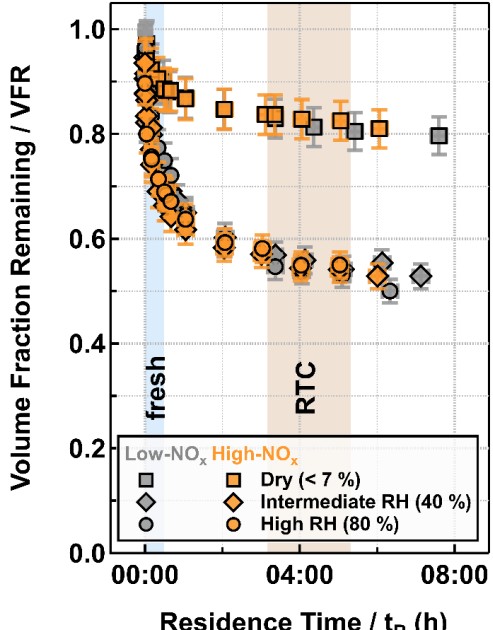

**Figure 1.** Evapograms for low-NO$_x$ (grey) and high-NO$_x$ (orange) cases under dry (< 7 %, squares), intermediate RH (40 % RH, diamonds) and high RH (80 % RH, circles) conditions. The blue and brown areas indicate the collection periods of FIGAERO-CIMS corresponding to fresh and RTC samples.



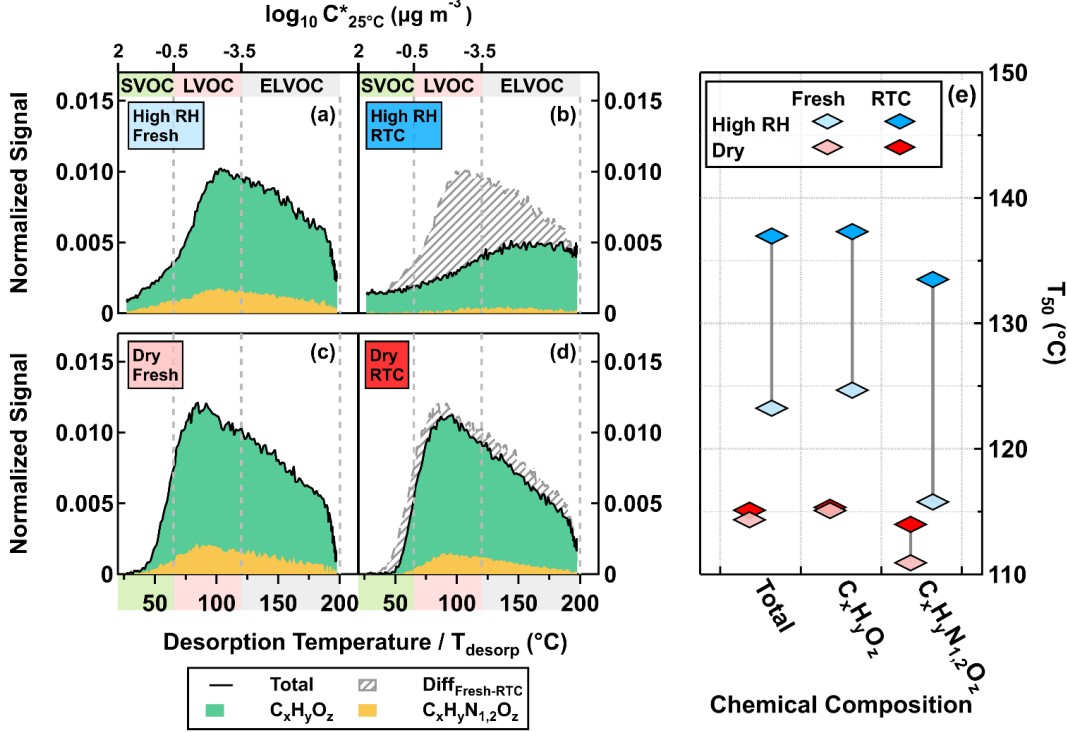


**Figure 2.** Sum thermograms (STG) (a – d) and median desorption temperature ($T_{50}$, diamonds) (e) for the high-NO$_x$ case under dry (RH < 7 %) and high RH (80 % RH) conditions. Non-nitrated organics and organic nitrates are indicated by $C_xH_yO_z$ and $C_xH_yN_{1,2}O_z$, respectively. On the panels (a – d), the solid black lines indicate the total signals of STGs with the green and yellow areas marking the contributions of $C_xH_yO_z$ and $C_xH_yN_{1,2}O_z$ to the STGs, respectively., The gray-striped areas represent
the differences in STGs between fresh and RTC stages. The color bands on the abscissa indicate volatility classes. Note that we presented the STGs of RTC stages after accounting for changes in the average VFR (VFR$_{avg}$) between fresh and RTC stages during the FIGAERO sample time.

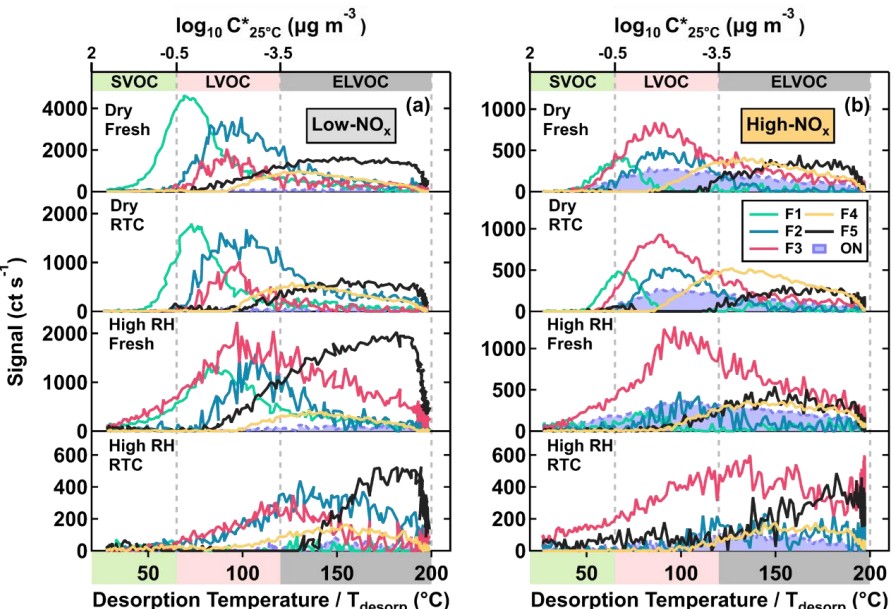

**Figure 3.** Factor thermograms of the five sample factors from the 12-factor PMF solution of non-nitrated organics in α-pinene SOA particles under low-NO$_x$ (a) and high-NO$_x$ (b) conditions. In addition, thermograms of organic nitrates (sum of NF1 − 3) are shown as purple areas in panel (b). In both panels, the ranges of different volatility classes are highlighted as color bands on the abscissa.





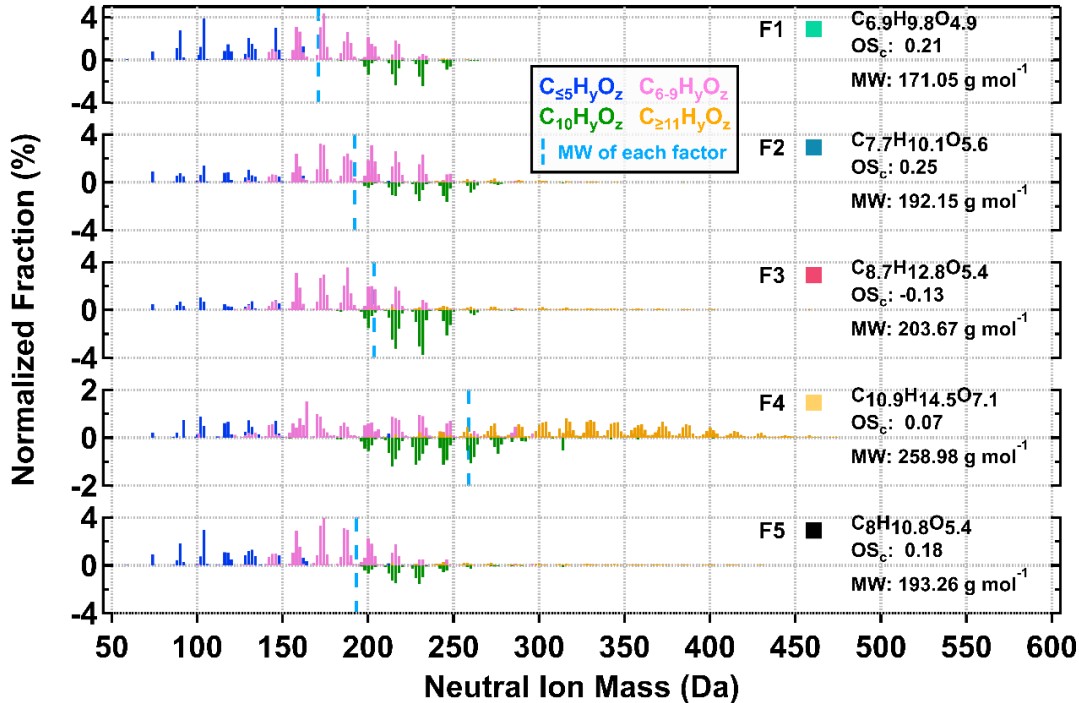


**Figure 4.** Normalized factor mass spectra of the five sample factors from the 12-factor PMF solution of non-nitrated organics in α-pinene SOA particles under two $NO_x$ conditions. For each factor mass spectrum, its signal-weighted molecular composition, molecular weight (MW), and oxidation state ($OS_c$) are shown on the right. Right next to each factor label, the squares are shown in the same color scheme as the factor thermograms in Figure 3 to indicate different sample factors. The
blue dashed line indicates the average MW of each factor.

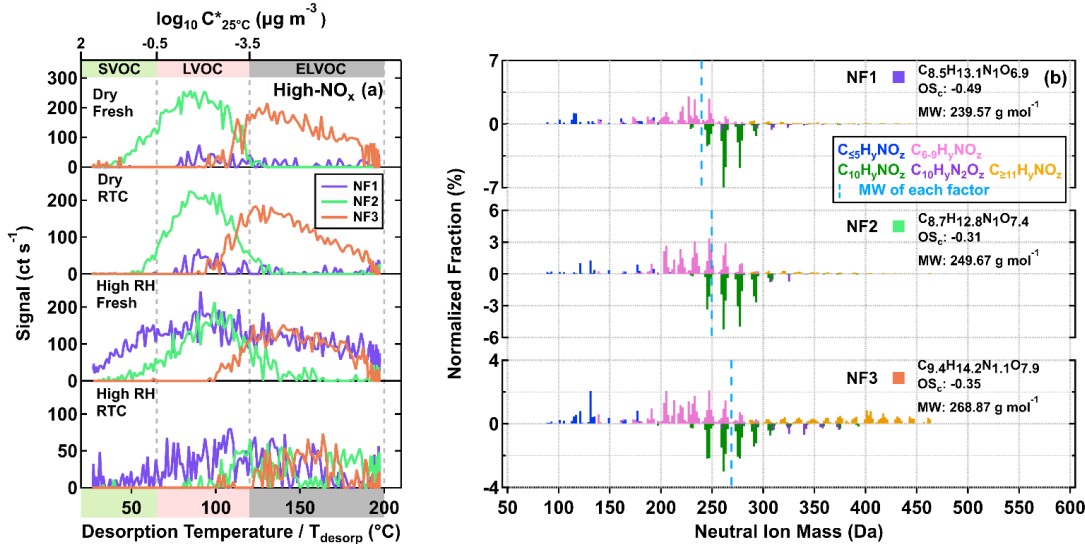

**Figure 5.** Factor thermograms (a) and normalized mass spectra (b) for the three sample factors from the eight-factor solution of organic nitrates in the α-pinene SOA particles under high-NO$_x$ conditions. In panel (a), the ranges of different volatility classes are indicated in color bands on the abscissa. In panel (b), the signal-weighted molecular composition, molecular weight (MW), and oxidation state (OS$_c$) for each factor are shown on the right. Right next to each factor label in panel (b), the squares are shown in the same color scheme as the factor thermograms in panel (a) to identify different sample factors. The blue dashed line indicates the average MW of each factor.





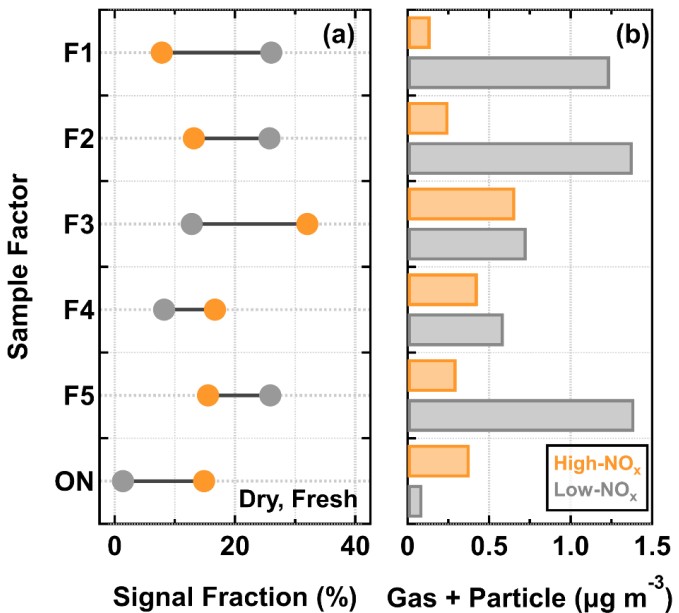


**Figure 6**. Signal contributions of sample factors to their total sum (a) and the corresponding estimated mass concentrations in gas and particle phase (b) for dry, fresh particles under low-$NO_x$ and high-$NO_x$ conditions. Note that here ON indicates the sum of the organic nitrate sample factors. The fraction is calculated by normalizing the measured sum from a PMF sample factor to the total sum of sample factors in the dry, fresh samples under low-$NO_x$ and high-$NO_x$ conditions.




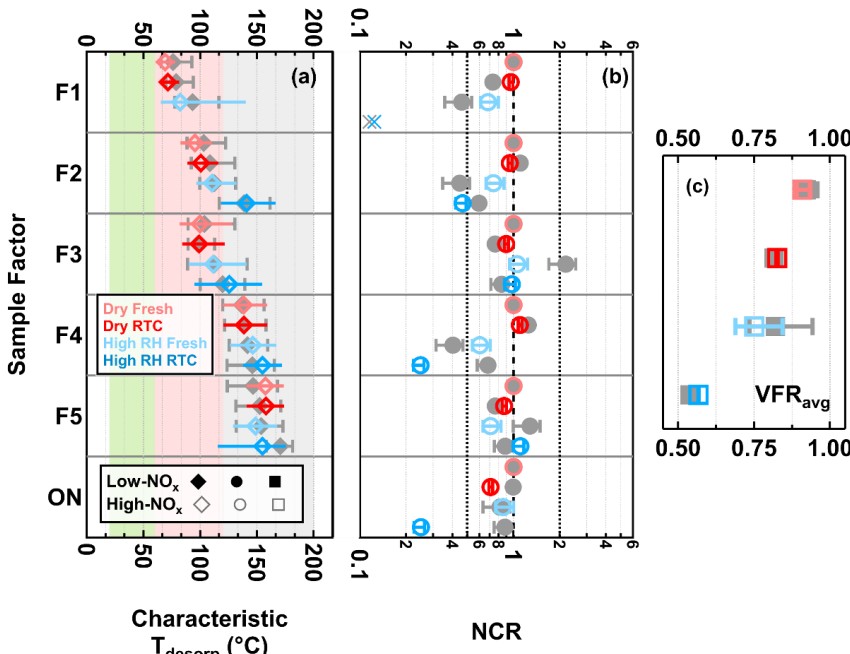

**Figure 7**. Volatility and changes in factor contribution for the five non-nitrated organics sample factors and total organic nitrates (ON, sum of organic nitrate sample factors). Panel (a): Characteristic desorption temperature (characteristic $T_{desorp}$). The marker indicates the $T_{50}$ values, and the horizontal lines mark the interquartile range of the factor thermograms. Panel (b): Net change ratio (NCR) with error bars indicating the uncertainty stemming from the estimated range in molecular weight and particle density. Panel c): average volume fraction remaining during sample collection (VFR$_{avg}$) with error bars indicating the minimum and maximum values. In all panels, the values for the high-NO$_x$ case are shown with colored markers while the low-NO$_x$ ones are displayed in grey. The colors indicate the sample type. The order of samples is identical for the low-NO$_x$ and high-NO$_x$ data. In panel (a), the range of volatility classes are highlighted with background colors (green – SVOCs; red – LVOCs; grey – ELVOCs). In panel (b), the dashed line at NCR equal to 1 indicates that any loss is counterbalanced by production, or no change occurs. The two dotted lines at NCR equal to 0.5 and 2 represent significant net loss and production.



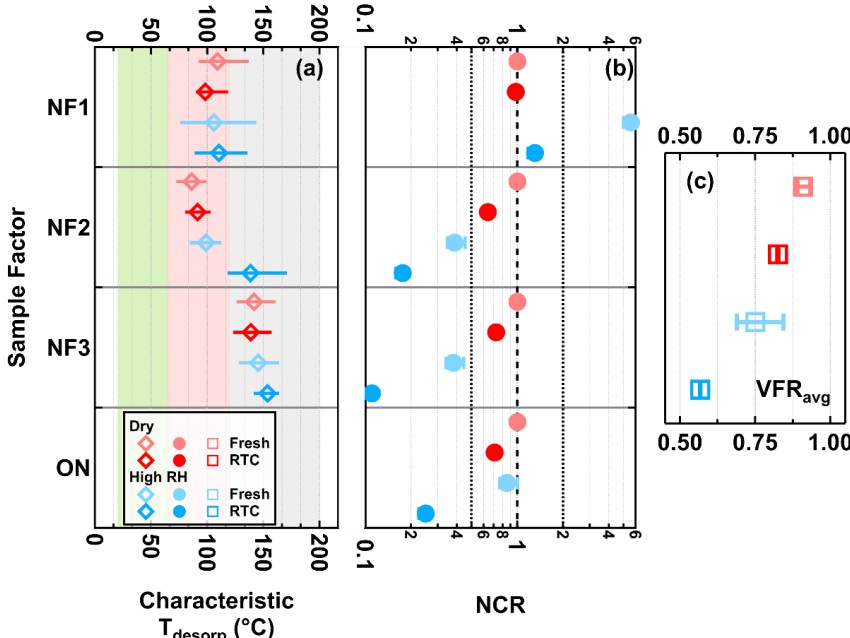

**Figure 8**. Volatility and changes in factor contribution for the three organic nitrate sample factors and total organic nitrates
(ON, sum of organic nitrate sample factors). Panel (a): Characteristic desorption temperature (characteristic $T_{desorp}$). The marker
indicates the $T_{50}$ values, and the horizontal lines mark the interquartile range of the factor thermograms. Panel (b): Net change
ratio (NCR) with error bars indicating the uncertainty stemming from the estimated range in molecular weight and particle
density. Panel c): average volume fraction remaining during sample collection (VFR$_{avg}$) with error bars indicating the minimum
and maximum values. The colors indicate the sample type. In panel (a), the range of volatility classes are highlighted with
background colors (green – SVOCs; red – LVOCs; grey – ELVOCs). In panel (b), the dashed line at NCR equal to 1 indicates
that any loss is counterbalanced by production, or no change occurs. The two dotted lines at NCR equal to 0.5 and 2 represent
significant net loss and production.