# Peer review of "Isothermal evaporation of $\alpha$ -pinene secondary organic aerosol particles formed under low-NOx and high-NOx conditions"

_Atmospheric Chemistry and Physics, 2022_

## Author Comment (AC1)

**Reply to Reviewer 1**

This study investigates the isothermal evaporation of  $\alpha$ -pinene secondary organic aerosol (SOA) formed under both lowand high-NOx conditions and under a range of relative humidity conditions. Applying positive matrix factorization (PMF) simplifies the analysis of mass spectra data. Linking the changes in individual PMF factors during isothermal evaporation and their volatility information obtained from FIGAERO-CIMS enables separation of the physical process - evaporation from chemical processes, e.g., hydrolysis. Although the evaporation behavior of  $\alpha$ -pinene SOA with low-NOx and the influence of humidity on particle evaporation behavior have been published in a previous paper (Li et al, 2021), I think it still provides valuable information on the evaporation and evolution of the SOA formed under high-NOx conditions. There are a few major and minor comments I would like the authors to address before it is considered for publication in ACP.

We would like to thank the reviewer for the positive feedback and valuable comments. We have improved the manuscript and presentation based on reviewer's suggestions. Below we give point-by-point response to the comments.

**Major comments:**

(1) A link between the PMF factors and their corresponding chemical reactions/pathways is missing. As the chemical composition of individual factors is available, it would be possible and great to build the link to help better understand the mechanisms behind the observed changes and differences.

Response: Thanks for the comment. Indeed, it would be very interesting if we could understand the chemical mechanisms underlying the changes in the PMF factors during isothermal evaporation especially at high RH. However, in practice, it could be extremely challenging due to the lack of details about the observed molecular compositions in general. The FIGAERO-CIMS only provides sum formulas for detected ions (i.e., their elemental composition) and not the molecular composition. That means we do not know the functionalities for each ion. Very likely, there could be isomeric compounds at one m/z. Additionally, the signals observed at high desorption temperature (e.g., 100 °C) might not be from the direct desorption from the compounds but instead from the decomposition of thermally labile compounds on the FIGERO filter. Very little information is obtainable for their parent compounds. Therefore, it is very unrealistic to propose any detailed chemical mechanism behind the observed changes and differences on the basis of very limited information about chemical compounds in the SOA particles.

(2) I am wondering about any relationship and/or correlation between the factors of non-nitrated organics and the factors of organic nitrates for high-NOx systems? I understand separating non-nitrated organics and organic nitrate allows having common PMF factors for low- and high-NOx systems. However, especially when discussing potential transformation, these factors are closely related. The discussion of NCR of the factors of non-nitrated organics and the factors of organic nitrates should not be separated.

Response: Thanks for the suggestion. We restructured the sections 3.4.1, 3.4.2 and 3.4.3. We now combined the discussion about the NCR for both non-nitrated organics (now labelled as "CHO compounds") and organic nitrates (labelled as ON). The section 3.4.3 is now entitled "Possible aqueous-phase processes affecting NCR of CHO compounds and ON". We also made a schematic diagram which is shown in the SI to summarize the processes of different samples factors under two high-RH conditions for the low-NOx and high NOx cases.

**[..]**

In the high-NOx samples, only decreasing or constant NCR values were observed for factors F2 - F5 at the fresh stage under high-RH conditions. The formation pathways for products grouping into F3 and F5 are minor since there was no increase in either of their NCR values. This additionally suggests that the reactions consuming F2 and F4 are mostly hydrolysis type processes and create products that do not remain in the particle phase. This formation of more volatile compounds may increase the observed isothermal evaporation of the particles. Under the same condition, NF2 and NF3 showed decreasing NCR values but NF1 exhibited an increase in its NCR. Even though the NCR of NF2 and NF3 also showed a decreasing trend with increasing evaporation at the RTC stage under low-RH conditions, we did not observe any production of compounds grouped into NF1 (Figure 6a) or an increase in its NCR (Error! Reference source not found.b). One explanation for the difference in the NCR of NF1 could be that the functional groups involved in the decomposition process vary between the two evaporation stages (low RH, RTC vs. high RH, fresh). As the ON produced in the OFR are multifunctional, hydrolysis is not necessarily restricted to the nitrate groups under high-RH conditions. Other functional groups like peroxides and esters can undergo hydrolysis as well. Compounds in NF1 could be products from the decomposition of NF2 and NF3. It is also possible that compounds in NF1 could be formed via the nitration of alcohols in the presence of HNO3 (Wang et al., 2021). While alcohols could be from CHO samples factors (e.g., F2, F4 and F5) that displayed decreasing NCR values, HNO3 could be produced from the decomposition of NF2 and NF3 or already be present in the particles from uptake during the SOA formation in the OFR under high-NOx conditions.

**[...]**

When particles continued to evaporate in the RTC under high-RH conditions, there were again differences in the evolution of F2 – F5 for low-NOx and high-NOx conditions. While F2 and F4 experienced increases in their NCR in the low-NOx samples, these two factors displayed continuous decreases in their NCR in the high-NOx ones. For the three ON factors in the high-NOx samples, they consistently showed decreasing NCR values, and also exhibited more significant decreases in their NCR values as compared with most of the CHO factors. As a majority of compounds in each ON factor were desorbed in the volatility range of LVOCs and ELVOCs, their decreasing NCR values were highly likely caused by aqueous-phase process instead of evaporation. [...] Figure S12 shows a graphical representation of the described processes occurring during the isothermal evaporation of  $\alpha$ -pinene SOA particles under two high-RH evaporation stages (fresh vs RTC).

Figure S12. Processes during the isothermal evaporation of  $\alpha$ -pinene SOA particles at two evaporation stage (fresh vs RTC) at high RH. Low-and high-NOx conditions indicate the levels of NOx during SOA formation in the OFR. Green and blue arrows suggest the loss pathways into gas phase due to direct evaporation and those due to evaporation after reactions, respectively. The yellow circles indicate the reaction intermediates formed in the aqueous-phase reactions that were shown in pink arrows. Pink arrows going into and out of the area of reactions intermediates represent aqueous-reactions consuming compounds of factors and those producing compounds of factors, respectively.

**Minor comments:**

Line 160: It would be nice to add some explanation about why the thermograms of factors (with fixed molecular composition, oxidation state, etc) would change/shift under different conditions.

Response: In our previous paper, we have provided detailed discussion on the changes in the thermograms of factors (Li et al., 2021) (see supplement S1.2.3 in Li et al. (2021)). In the context of this manuscript, we only provide a brief discussion about changes in the thermogram shapes. PMF groups the compounds which follows the most similar temporal behavior (i.e., desorption behavior on the FIGAERO and/or aqueous phase process) during isothermal evaporation into

factors. The PMF result can be seen as a compromise between finding as few as factors as possible and reconstructing the majority of the ion thermograms correctly.

For the case of F1, PMF finds the compounds or signals which have most similar volatility. During isothermal evaporation, the most volatile fraction of the factor F1 had more evaporation than the other fraction. We would ideally allow adjustments in the mass spectrum of F1 to account for these changes, but PMF does not allow variations in any factor mass spectrum. Therefore, the molecular composition and oxidation state are fixed. In order to minimize the residuals for all ions present in the factor, PMF finds an optimal compromise by allowing a slight increase in the characteristic  $T_{desorp}$ .

For the second type of changes in the thermograms that were showed by e.g., F3 in the low-NOx case, the change in the thermograms between low-RH and high-RH samples under fresh conditions cannot be explained without considering particle-phase chemistry. The compounds in F3 are in the volatility range of LVOC and thus most of them are expected to remain in the particle phase within the evaporation time of 0.5 hours. Thus, isothermal evaporation cannot explain any changes in this factor in these samples. The key is to realize that FIGAERO CIMS measurements detects the sum formulas of ions of compounds that are volatilized from the FIGAERO filter. An ion A may stem from a compound A or its isomer A'. Or a thermally instable compound with low volatility decomposed upon heating and the resulting compound A is detected. For the grouping of PMF factors, the correlations between ion signals are important. For that correlation the "x-values" are irrelevant in the algorithm. I.e., Tdesorp values are not used in the PMF algorithm. As long as the relation between the signals of a set of ion signals is the same in multiple samples, they will be grouped together.

Let us assume that there are the compounds A, B, and C in the fresh low-RH sample, and they are detected by FIGAERO CIMS as the ions A, B, and C grouped into one factor. In the fresh, high-RH sample, ions with the same correlations to each other are found, but they may stem from isomeric compounds A', B', and C' which have a higher volatility and thus apparently increase the contribution of SVOC compounds for that factor. A', B', and C' could be formed from aqueous phase reactions, e.g., hydrolysis of oligomers. In addition, the increased contributions of ELVOC compounds can be caused by the formation of new oligomers (e.g., A-B, A-A, B-C) which involved the monomer units A, B and C in aqueous phase. Such dimers are supposed to have lower volatility than that of the monomer units, but their coupling bonds between monomer units might be very vulnerable against high temperature. That means that very likely, before these compounds could thermally desorb from the FIGAERO filter, the coupling bond will break so that original monomers could be released. The detected ions in FIGAERO-CIMS would again show up as A, B, C in the same ratio as the dry sample. Since now we observe thermal decomposition of the dimers instead of direct desorption of the monomers, the Tdesrop would have shifted to higher values and the shape of the thermogram would have changed.

Line 267-268: For F5,  $T_{max}$  is actually the temperature of thermal decomposition, right? In this case, how did you calculate the volatility based on the thermal decomposition temperature?

Response: For factors dominated by thermal decomposition, the interpretation of  $T_{max}$  as a physical property becomes difficult. Assuming an Arrhenius-style temperature dependence, the decomposition is enhanced when the temperature increases, and more signal is detected for the decomposition products. The position of the maximum of the factor thermogram then depends on the balance between that increase and the availability of the decomposing compounds. As the increase is strongly exponential, the  $T_{max}$  will mainly depend on the kinetics of the decomposition reaction (e.g., activation energy and pre-exponential factor). If the decomposing compounds also have sources (e.g., through other thermally enhanced reactions), matters complicate. Thus, the  $T_{max}$  or characteristic  $T_{desorp}$  values only provide a general idea about the temperature range in which the thermal decomposition of the compounds grouped into the factor occurs.

The thermal decomposition is happening at temperatures below the (theoretical) desorption of the parent compounds. Thus, using the characteristic  $T_{desorp}$  of F5 we can only obtain an "apparent" volatility. The true volatility of the intact parent compound will be lower than that. We have now discussed this briefly in the manuscript.

Change: Section 3.2.1 During a FIGAERO desorption cycle, a thermally labile compound starts to decompose into smaller compounds at desorption temperatures that are above its threshold temperature of decomposition. When the desorption temperature increases, thermal decomposition is enhanced, and more signal is detected for the decomposition products. For F5 specifically, the position of its  $T_{50}$  depends on the balance between the increasing decomposition and the availability of the decomposing compounds. Thus, the  $T_{50}$  value only provides a general idea about the temperature range in which the thermal decomposition of the compounds grouped into the F5 occurs. The thermal decomposition very likely happens at temperatures below the (theoretical) desorption temperature of the parent compounds. Thus, we can only obtain an "apparent" volatility with the use of the  $T_{50}$  value of F5.

Line 320 -322: This sentence is difficult to follow.

Response: We modified the sentence to enhance readability.

Change: Section 3.4.1 [...] With increasing evaporation time at high RH, the evolution pattern of the NCR for either F2 or F4 differed between two SOA types. [...]

Line 386 – 398: In Figure 7, there is no NCR for individual NF factors, I would suggest adding the NCR of individual NFs to Figure 7 or a SI figure.

Response: The NCR of the NF factors was previously shown in Figure 8. Following the suggestion of the reviewer we now combined these two figures in the manuscript.

---

## Author Comment (AC2)

**Reply to Reviewer 1**

This study investigates the isothermal evaporation of α-pinene secondary organic aerosol (SOA) formed under both low- and high-NOx conditions and under a range of relative humidity conditions. Applying positive matrix factorization (PMF) simplifies the analysis of mass spectra data. Linking the changes in individual PMF factors during isothermal evaporation and their volatility information obtained from FIGAERO-CIMS enables separation of the physical process - evaporation from chemical processes, e.g., hydrolysis. Although the evaporation behavior of α-pinene SOA with low-NOx and the influence of humidity on particle evaporation behavior have been published in a previous paper (Li et al, 2021), I think it still provides valuable information on the evaporation and evolution of the SOA formed under high-NOx conditions. There are a few major and minor comments I would like the authors to address before it is considered for publication in ACP.

We would like to thank the reviewer for the positive feedback and valuable comments. We have improved the manuscript and presentation based on reviewer's suggestions. Below we give point-by-point response to the comments.

**Major comments:**

(1) A link between the PMF factors and their corresponding chemical reactions/pathways is missing. As the chemical composition of individual factors is available, it would be possible and great to build the link to help better understand the mechanisms behind the observed changes and differences.

Response: Thanks for the comment. Indeed, it would be very interesting if we could understand the chemical mechanisms underlying the changes in the PMF factors during isothermal evaporation especially at high RH. However, in practice, it could be extremely challenging due to the lack of details about the observed molecular compositions in general. The FIGAERO-CIMS only provides sum formulas for detected ions (i.e., their elemental composition) and not the molecular composition. That means we do not know the functionalities for each ion. Very likely, there could be isomeric compounds at one m/z. Additionally, the signals observed at high desorption temperature (e.g., 100 °C) might not be from the direct desorption from the compounds but instead from the decomposition of thermally labile compounds on the FIGERO filter. Very little information is obtainable for their parent compounds. Therefore, it is very unrealistic to propose any detailed chemical mechanism behind the observed changes and differences on the basis of very limited information about chemical compounds in the SOA particles.

(2) I am wondering about any relationship and/or correlation between the factors of non-nitrated organics and the factors of organic nitrates for high-NOx systems? I understand separating non-nitrated organics and organic nitrate allows having common PMF factors for low- and high-NOx systems. However, especially when discussing potential transformation, these factors are closely related. The discussion of NCR of the factors of non-nitrated organics and the factors of organic nitrates should not be separated.

Response: Thanks for the suggestion. We restructured the sections 3.4.1, 3.4.2 and 3.4.3. We now combined the discussion about the NCR for both non-nitrated organics (now labelled as "CHO compounds") and organic nitrates (labelled as ON). The section 3.4.3 is now entitled "Possible aqueous-phase processes affecting NCR of CHO compounds and ON". We also made a schematic diagram which is shown in the SI to summarize the processes of different samples factors under two high-RH conditions for the low-$NO_x$ and high $NO_x$ cases.

Change: Section 3.4.3

[..]

In the high-NO$_x$ samples, only decreasing or constant NCR values were observed for factors F2 – F5 at the fresh stage under high-RH conditions. The formation pathways for products grouping into F3 and F5 are minor since there was no increase in either of their NCR values. This additionally suggests that the reactions consuming F2 and F4 are mostly hydrolysis type processes and create products that do not remain in the particle phase. This formation of more volatile compounds may increase the observed isothermal evaporation of the particles. Under the same condition, NF2 and NF3 showed decreasing NCR values but NF1 exhibited an increase in its NCR. Even though the NCR of NF2 and NF3 also showed a decreasing trend with increasing evaporation at the RTC stage under low-RH conditions, we did not observe any production of compounds grouped into NF1 (Figure 6a) or an increase in its NCR (**Error! Reference source not found.**b). One explanation for the difference in the NCR of NF1 could be that the functional groups involved in the decomposition process vary between the two evaporation stages (low RH, RTC vs. high RH, fresh). As the ON produced in the OFR are multifunctional, hydrolysis is not necessarily restricted to the nitrate groups under high-RH conditions. Other functional groups like peroxides and esters can undergo hydrolysis as well. Compounds in NF1 could be products from the decomposition of NF2 and NF3. It is also possible that compounds in NF1 could be formed via the nitration of alcohols in the presence of HNO$_3$ (Wang et al., 2021). While alcohols could be from CHO samples factors (e.g., F2, F4 and F5) that displayed decreasing NCR values, HNO$_3$ could be produced from the decomposition of NF2 and NF3 or already be present in the particles from uptake during the SOA formation in the OFR under high-NO$_x$ conditions.

[…]

When particles continued to evaporate in the RTC under high-RH conditions, there were again differences in the evolution of F2 – F5 for low-NO$_x$ and high-NO$_x$ conditions. While F2 and F4 experienced increases in their NCR in the low-NO$_x$ samples, these two factors displayed continuous decreases in their NCR in the high-NO$_x$ ones. For the three ON factors in the high-NO$_x$ samples, they consistently showed decreasing NCR values, and also exhibited more significant decreases in their NCR values as compared with most of the CHO factors. As a majority of compounds in each ON factor were desorbed in the volatility range of LVOCs and ELVOCs, their decreasing NCR values were highly likely caused by aqueous-phase process instead of evaporation. […] Figure S12 shows a graphical representation of the described processes occurring during the isothermal evaporation of α-pinene SOA particles under two high-RH evaporation stages (fresh vs RTC).

[Figure]

Figure S12. Processes during the isothermal evaporation of α-pinene SOA particles at two evaporation stage (fresh vs RTC) at high RH. Low-and high-NO$_x$ conditions indicate the levels of NO$_x$ during SOA formation in the OFR. Green and blue arrows suggest the loss pathways into gas phase due to direct evaporation and those due to evaporation after reactions, respectively. The yellow circles indicate the reaction intermediates formed in the aqueous-phase reactions that were shown in pink arrows. Pink arrows going into and out of the area of reactions intermediates represent aqueous-reactions consuming compounds of factors and those producing compounds of factors, respectively.

**Minor comments:**

Line 160: It would be nice to add some explanation about why the thermograms of factors (with fixed molecular composition, oxidation state, etc) would change/shift under different conditions.

Response: In our previous paper, we have provided detailed discussion on the changes in the thermograms of factors (Li et al., 2021) (see supplement S1.2.3 in Li et al. (2021)). In the context of this manuscript, we only provide a brief discussion about changes in the thermogram shapes. PMF groups the compounds which follows the most similar temporal behavior (i.e., desorption behavior on the FIGAERO and/or aqueous phase process) during isothermal evaporation into

factors. The PMF result can be seen as a compromise between finding as few as factors as possible and reconstructing the majority of the ion thermograms correctly.

For the case of F1, PMF finds the compounds or signals which have most similar volatility. During isothermal evaporation, the most volatile fraction of the factor F1 had more evaporation than the other fraction. We would ideally allow adjustments in the mass spectrum of F1 to account for these changes, but PMF does not allow variations in any factor mass spectrum. Therefore, the molecular composition and oxidation state are fixed. In order to minimize the residuals for all ions present in the factor, PMF finds an optimal compromise by allowing a slight increase in the characteristic $T_{desorp}$.

For the second type of changes in the thermograms that were showed by e.g., F3 in the low-$NO_x$ case, the change in the thermograms between low-RH and high-RH samples under fresh conditions cannot be explained without considering particle-phase chemistry. The compounds in F3 are in the volatility range of LVOC and thus most of them are expected to remain in the particle phase within the evaporation time of 0.5 hours. Thus, isothermal evaporation cannot explain any changes in this factor in these samples. The key is to realize that FIGAERO CIMS measurements detects the sum formulas of ions of compounds that are volatilized from the FIGAERO filter. An ion A may stem from a compound A or its isomer A'. Or a thermally instable compound with low volatility decomposed upon heating and the resulting compound A is detected. For the grouping of PMF factors, the correlations between ion signals are important. For that correlation the "x-values" are irrelevant in the algorithm. I.e., $T_{desorp}$ values are not used in the PMF algorithm. As long as the relation between the signals of a set of ion signals is the same in multiple samples, they will be grouped together.

Let us assume that there are the compounds A, B, and C in the fresh low-RH sample, and they are detected by FIGAERO CIMS as the ions A, B, and C grouped into one factor. In the fresh, high-RH sample, ions with the same correlations to each other are found, but they may stem from isomeric compounds A', B', and C' which have a higher volatility and thus apparently increase the contribution of SVOC compounds for that factor. A', B', and C' could be formed from aqueous phase reactions, e.g., hydrolysis of oligomers. In addition, the increased contributions of ELVOC compounds can be caused by the formation of new oligomers (e.g., A-B, A-A, B-C) which involved the monomer units A, B and C in aqueous phase. Such dimers are supposed to have lower volatility than that of the monomer units, but their coupling bonds between monomer units might be very vulnerable against high temperature. That means that very likely, before these compounds could thermally desorb from the FIGAERO filter, the coupling bond will break so that original monomers could be released. The detected ions in FIGAERO-CIMS would again show up as A, B, C in the same ratio as the dry sample. Since now we observe thermal decomposition of the dimers instead of direct desorption of the monomers, the $T_{desorp}$ would have shifted to higher values and the shape of the thermogram would have changed.

Line 267-268: For F5, $T_{max}$ is actually the temperature of thermal decomposition, right? In this case, how did you calculate the volatility based on the thermal decomposition temperature?

Response: For factors dominated by thermal decomposition, the interpretation of $T_{max}$ as a physical property becomes difficult. Assuming an Arrhenius-style temperature dependence, the decomposition is enhanced when the temperature increases, and more signal is detected for the decomposition products. The position of the maximum of the factor thermogram then depends on the balance between that increase and the availability of the decomposing compounds. As the increase is strongly exponential, the $T_{max}$ will mainly depend on the kinetics of the decomposition reaction (e.g., activation energy and pre-exponential factor). If the decomposing compounds also have sources (e.g., through other thermally enhanced reactions), matters complicate. Thus, the $T_{max}$ or characteristic $T_{desorp}$ values only provide a general idea about the temperature range in which the thermal decomposition of the compounds grouped into the factor occurs.

The thermal decomposition is happening at temperatures below the (theoretical) desorption of the parent compounds. Thus, using the characteristic $T_{desorp}$ of F5 we can only obtain an "apparent" volatility. The true volatility of the intact parent compound will be lower than that. We have now discussed this briefly in the manuscript.

Change: Section 3.2.1    During a FIGAERO desorption cycle, a thermally labile compound starts to decompose into smaller compounds at desorption temperatures that are above its threshold temperature of decomposition. When the desorption temperature increases, thermal decomposition is enhanced, and more signal is detected for the decomposition products. For F5 specifically, the position of its $T_{50}$ depends on the balance between the increasing decomposition and the availability of the decomposing compounds. Thus, the $T_{50}$ value only provides a general idea about the temperature range in which the thermal decomposition of the compounds grouped into the F5 occurs. The thermal decomposition very likely happens at temperatures below the (theoretical) desorption temperature of the parent compounds. Thus, we can only obtain an "apparent" volatility with the use of the $T_{50}$ value of F5.

Line 320 -322: This sentence is difficult to follow.

Response: We modified the sentence to enhance readability.

Change: Section 3.4.1    […] With increasing evaporation time at high RH, the evolution pattern of the NCR for either F2 or F4 differed between two SOA types. […]

Line 386 – 398: In Figure 7, there is no NCR for individual NF factors, I would suggest adding the NCR of individual NFs to Figure 7 or a SI figure.

Response: The NCR of the NF factors was previously shown in Figure 8. Following the suggestion of the reviewer we now combined these two figures in the manuscript.

Change:

[Figure]

**Figure 1.** Volatility and changes in factor contribution for the five CHO and three ON sample factors and total ON (sum of ON sample factors). Panel (a): Characteristic desorption temperature (characteristic $T_{desorp}$). The marker indicates the $T_{50}$ values, and the horizontal lines mark the interquartile range of the factor thermograms. Panel (b): Net change ratio (NCR) with error bars indicating the uncertainty stemming from the estimated range in molecular weight and particle density. Panel (c): average volume fraction remaining during sample collection ($VFR_{avg}$) with error bars indicating the minimum and maximum values. In all panels, the values for the high-$NO_x$ case are shown with colored markers while the low-$NO_x$ ones are displayed in grey. The colors indicate the sample type. The order of samples is identical for the low-$NO_x$ and high-$NO_x$ data. In panel (a), the range of volatility classes are highlighted with background colors. In panel (b), the dashed line at NCR equal to 1 indicates that any loss is counterbalanced by production, or no change occurs. The two dotted lines at NCR equal to 0.5 and 2 represent significant net loss and production. For F1, its factor thermogram contributes less than 5% to the total sample signal and does not show a clear maximum under the high-RH and RTC condition. Therefore, its characteristic $T_{desorp}$ value was not estimated in panel (a) and its NCR was be indicated by a cross close to the 0.1 in panel (b).

Line 682: "FIGERO" -> "FIGAERO".

Response: We changed it accordingly.

**SI**

Figure S6, S10: what are "DMA blanks" and "snap blanks"? Specify them.

Response: Thanks for pointing that out. Now we renamed the snap blank as zero blank for clarity. We added the description about "DMA blanks" and "zero blanks" in the caption of Figure S6 (now Figure S7) but also in that of Figure S4 (now Figure S5).

Change:

**Figure S5**. […] DMA blanks were determined by analyzing samples that were collected through the NanoDMAs with a set voltage of 0 V (i.e., no selected particles) for 0.5 hours. Zero blanks were investigated by measuring FIGAERO filters with no particle collection. […]

**Figure S7**. […] DMA blanks were determined by analyzing samples that were collected through the NanoDMAs with a set voltage of 0 V (i.e., no selected particles) for 0.5 hours. Zero blanks were conducted by measuring FIGAERO filters with no particle collection. […]

Figure S6: In the figures of high-NOx dry fresh and dry RTC, the background factor (black dashed lines) has nice thermograms with Tmax of 60 – 70? It is different compared to their thermograms in blanks (e.g. Snap blank 40). Why?

Response: We are aware that for this factor, the shapes of thermograms in the two dry samples were different from those in the high-RH RTC and snap blank 40 blanks and do look more like a sample factor than a background factor. We have explored multiple PMF solutions and their rotations, but all solutions have this feature.

It seems tempting to assign this factor as a sample factor rather than a background factor when just looking at the shapes of thermograms in the high-NO$_x$ dry fresh and dry RTC samples. However, we decided to assign this factor as a background factor after thoroughly considering the following two reasons:

1) Although this factor has "nice-looking" thermograms, it has 90% lower signal contributions to the dry particle samples compared with each of the two major sample factors that are in green and orange.

2) Compared with the two dry particle samples, this factor showed similar and even higher absolute signal intensities in the high-RH RTC and zero blank 40 samples which had lower mass loadings. Additionally, the factor did not show any clear thermogram shape in the high-RH and snap blank 40 samples.

**Reply to Reviewer 2**

Li et al. measured the humidity-dependent evaporation rate of SOA particles generated from the photooxidation of alpha-pinene in the presence and absence of added $NO_x$. Detailed chemical composition and volatility information was obtained using a FIGAERO-CIMS, and positive matrix factorization was applied to identify volatility-resolved classes of oxidation products, including organic nitrates. This study is an extension of similar/previous work performed by the Kuopio research group. I would support eventual publication after consideration of my comments below.

We would like to thank Dr. Lambe for his constructive and insightful comments. We have now improved the manuscript based on them. Below we give point-by-point responses to the comments.

1. The authors examine correlations between VFR in the residence time chamber and FIGAERO-CIMS characteristic thermal desorption temperature ($T_{50}$). Ultimately, only the FIGAERO-CIMS thermograms were used to infer the SOA volatility distributions. These SOA volatility distributions can likewise be derived from the evapograms and compared/contrasted with thermogram-derived volatility distributions, as has been done previously by this group (Tikkanen et al, 2020). In my opinion, a similar analysis should be done here. While adding $NO_x$ to generate organic nitrates is certainly a novel component of this study, a significant portion of the results was dedicated to the analysis of the isothermal evaporation of alpha-pinene SOA under low-$NO_x$ conditions, which has already been published in various forms (e.g., Buchholz et al, 2019; Li et al., 2021). Comparing C* distributions obtained via evapograms and thermograms is a logical addition to this paper that will increase its impact by investigating the utility of evapograms as inputs to chemistry and climate models.

   Response: We totally agreed that it will be interesting to compare the volatility distributions derived from the evapograms with that estimated from the FIGAERO-CIMS thermograms. However, to conduct such comprehensive analysis would require huge amounts of time and efforts in setting up the model especially for the highly oxidized samples in this manuscript. Since aqueous phase reactions have been observed in the samples during isothermal evaporation, the model needs to account for such reactions that are still underexplored. Additionally, compounds (e.g., peroxide) that can be reactants or products in the aqueous phase reaction might undergo thermal decomposition on the FIGAERO-CIMS. Overall, incomplete information about the aqueous phase reactions and particulate compositions would lead to considerable uncertainty in the results.

   Although the isothermal evaporation of α-pinene SOA particles formed under low-$NO_x$ conditions has been overlapped with the part of focuses in (Buchholz et al., 2019; Li et al., 2021), it is still worthwhile to investigate the low-$NO_x$ case. The main focus of our study is to understand how adding $NO_x$ during SOA formation affects particle composition and particle evaporation behavior. Without the low-$NO_x$ as the reference case, it is impossible to draw any conclusion about the effect of $NO_x$ on the composition by only running the PMF analysis for the high-$NO_x$ case. Therefore, it is necessary to include the low-$NO_x$ case in our study. To add more emphasizes to the role of organic nitrates and the high $NO_x$ case (and following the suggestion of reviewer #1), we restructured the NCR discussion and interpretation.

2. **L88**: This is not an accurate description of the $N_2O$-based photochemistry that occurs in the OFR254-i$N_2O$ mode because $N_2O$ does not photolyze significantly at 254 nm. Rather, NO and $NO_2$ are generated from the reaction: $N_2O + O(^1D) \rightarrow 2NO$ and $NO + O_3 \rightarrow NO_2 + O_2$. In OFR185-i$N_2O$, however, note that $N_2O$ photolysis at $\lambda = 185$ nm can generate NO via $N_2O + hv185 \rightarrow N_2 + O(^1D)$ followed by $N_2O + O(^1D) \rightarrow 2NO$.

Response: Thank you for pointing out our oversimplification in describing the OFR chemistry. We corrected the description of the $N_2O$-based photochemistry and also added the reactions for the OFR254-i$N_2O$ mode. We realize now that the oxidation system is too complex for the simple OFR OH exposure estimator and use the more sophisticated KinSim Model for calculating oxidants exposures and branching ratios (see also response to comment #4).

Change: Section 2.1 […] In the high-$NO_x$ case, $N_2O$ (99.5 % purity, mixing ratio inside OFR: 1.85 % in volume) were injected into the OFR for $NO_x$ production. NO and $NO_2$ were produced via the reactions $N_2O + O(^1D)$ →2NO and $NO + O_3 \rightarrow NO_2 + O_2$ under the illumination of the same 254-nm lamps inside the OFR (Lambe et al., 2017). […]

3. **L90**: The OH exposure should be in units of molecules $cm^{-3}$ s, not molecules $cm^{-3}$.

   Response: We corrected the unit accordingly.

4. **L90-L92**: $NO_3$ radicals are also generated using the OFR254-i$N_2O$ method ($NO_2+O_3\rightarrow NO_3+ O_2$). Please add the appropriate $NO_3$-based reactions that are listed in Palm et al. (2017) or Lambe et al. (2020) to the photochemical box model that was (presumably) used here. Then, calculate the $NO_3$ exposure values over the range of OFR254-i$N_2O$ conditions that were studied, and report the fractional oxidative loss of α-pinene to OH, $O_3$, and $NO_3$. For example, the fractional loss of alphα-pinene due to reaction with OH would be: f_aPinene_OH $= k_{OH}*OH_{exp} / (k_{OH}*OH_{exp} + k_{O3}*O_{3exp} + k_{NO3}*NO_{3exp})$. Was $IN_2O_5^-$ observed in the gas-phase iodide-CIMS spectra that were obtained for "fresh" samples?

   Response: Thank you for the comment. Originally, we only used the simplified OFR OH Exposure Estimator (v3.1). We realize now that that estimator is a bit too simplistic for our experiments and decided to use the KinSim model applying the OFR chemistry template for the simulation of the oxidation conditions in the OFR (Peng and Jimenez, 2019). The mechanism in the template includes the $NO_3$-based reactions that are listed in Palm et al. (2017). In addition, we also included reactions for the $RO_2$ fate simulations (Peng et al., 2019), shown in table R1.

   Using the KinSim model, we recalculated the OH exposure, and also estimated the $NO_3$ exposure and $O_3$ exposure. Then we calculated the fractional oxidation loss of α-pinene to OH, $O_3$ and $NO_3$. The fractional loss of α-pinene to OH, $O_3$, and $NO_3$ are 96%, 4%, and 0% in the low-$NO_x$ samples and 50%, 1%, and 49% for the high-$NO_x$ samples. We accordingly updated the information presented in Table S1 and adjusted the text in Section 2.1.

   Note that the values of the OH exposure change by a factor of 7 when using the more sophisticated model instead of the simple estimator. But this change is uniform for all conducted experiments and simply means that the overall OH exposure was higher than reported in the original manuscript.

   Unfortunately, the high concentrations of gaseous $HNO_3$ led to a severe depletion of the primary ion (I$^-$). I.e., we could not be sure if the analyte molecules would be ionized by I$^-$, $IHNO_3^-$ or $NO_3^-$. As sensitivity towards different functional groups strongly depends on the ionization scheme (e.g., I$^-$ vs. $NO_3^-$) and the ratio between the ions now depended on the incoming $HNO_3$ concentration. In other words, the FIGAERO-CIMS gas-phase measurements were biased by the very high gas-phase concentration of $HNO_3$. Therefore, we did not use the

FIGAERO-CIMS gas-phase measurements and the gas-phase $N_2O_5$ could not be interpreted. (see also response to comment #12).

**Table R1**. List of reactions and parameters used in the OFR chemistry modelling in KinSim. The rate constants are calculated with the use of the modified Arrhenius equation $k = A \cdot (\frac{T}{300})^{-n} \cdot e^{-\frac{E}{RT}}$. Parameters listed in the following table are the same as those used in Palm et al. (2017) and Peng et al. (2019) or those list in Tables B.9 and B.10 in Pandis and Seinfeld (2006).

| Reactant 1 | Reactant 2 | Product 1 | Product 2 | A_Ainf x $10^{12}$ | E_Einf | n_ninf | A0 | E0 | n0 |
|---|---|---|---|---|---|---|---|---|---|
| α-pinene | OH | $RO_2$ | $H_2O$ | 12.1 | -444 | 0 | 0 | 0 | 0 |
| α-pinene | $O_3$ | Product_$O_3$ | 0.9 OH | 1.01E-3 | 732 | 0 | 0 | 0 | 0 |
| α-pinene | $NO_3$ | $RO_2$ | | 1.19 | -490 | 0 | 0 | 0 | 0 |
| $RO_2$ | $NO_3$ | RO | | 1.50 | 0 | 0 | 0 | 0 | 0 |
| $RO_2$ | NO | RO | $NO_2$ | 9.00 | 0 | 0 | 0 | 0 | 0 |
| $RO_2$ | $HO_2$ | ROOH | $O_2$ | 15.00 | 0 | 0 | 0 | 0 | 0 |
| $RO_2$ | OH | $RPO_2$ | $H_2O$ | 100.00 | 0 | 0 | 0 | 0 | 0 |
| $RO_2$ | $RO_2$ | ROOR | | 0.10 | 0 | 0 | 0 | 0 | 0 |
| $RO_2$ | $O_3$ | RO | | 1.00E-5 | 0 | 0 | 0 | 0 | 0 |
| $RO_2$ | $NO_2$ | $RO_2NO_2$ | | 7.00 | 0 | 0 | 0 | 0 | 0 |

Change: Section 2.1 […] We used the KinSim model (Peng and Jimenez, 2019) and additionally implemented the reactions of $RO_2$ (Peng et al., 2019) for simulating the OFR chemistry. The OH exposure was estimated to be $(1.82 \pm 0.21) \times 10^{12}$ and $(2.45 \pm 0.09) \times 10^{12}$ molecules cm$^{-3}$ s for low-$NO_x$ and high-$NO_x$ conditions. Under low-$NO_x$ condition, OH and $O_3$ contributed to 96% and 4% of the loss of α-pinene. Under high-$NO_x$ condition, $NO_3$ radicals were produced from the $NO_2+O_3 \rightarrow NO_3+ O_2$ reaction and also contribute to the oxidation of α-pinene. The fractional loss of α-pinene to OH, $O_3$, and $NO_3$ were 50%, 1% and 49%. For the high-$NO_x$ case, the estimated ratio between the $[RO_2]$ + $[NO]$ and $[RO_2]$ + $[HO_2]$ pathways ($\frac{[RO_2]+[NO]}{[RO_2]+[HO_2]}$) was $0.84 \pm 0.19$. […]

SI

**Table S1**. Summary of experimental conditions and results of α-pinene SOA generation

| | Low-NO$_x$ | High-NO$_x$ |
|---|---|---|
| [VOC]$_{OFR}$ (ppb)[a] | 254 ± 11 | 296 ± 14 |
| [N$_2$O]$_{OFR}$ (%) | N/A | 1.82 ± 0.10 |
| [O$_3$]$_{OFR}$ (ppm)[b] | 9.76 ± 0.31 | 6.85 ± 0.36 |
| T$_{OFR}$ (°C) | 24.66 ± 0.76 | 28.14 ± 0.91 |
| RH$_{OFR}$ (%) | 44.19 ± 2.17 | 38.74 ± 2.63 |
| Nominal residence time (s) | 160 | 160 |
| effective OH exposure (10$^{12}$ molec cm$^{-3}$ s)[c] | 1.82 ± 0.21 | 2.45 ± 0.09 |
| $\dfrac{[RO_2] + [NO]}{[RO_2] + [HO_2]}$ | N/A | 0.84 ± 0.19 |
| fraction Loss to OH (%) | 96 | 50 |
| fraction Loss to O$_3$ (%) | 4 | 1 |
| fraction Loss to NO$_3$ (%) | 0 | 49 |
| oxygen-to-carbon (O:C)[d] | 0.77 ± 0.03 | 0.74 ± 0.01 |
| oxidation state (OS$_c$)[d] | 0.05 ± 0.04 | 0.02 ± 0.02 |

[a] Mixing ratio of α-pinene was corrected with the dilution of O$_3$-contained flow but without the loss due to pure ozonolysis at the inlet. [b] O$_3$ was measured at the OFR outlet after 254-nm UV lamps were switched on but without the addition of α-pinene and N$_2$O. [c] OH exposure was calculated with the KinSim model (Peng and Jimenez, 2019). [d] The values of the oxygen to carbon ratio (O:C) and the oxidation state (OS$_c$) were derived from the HR-ToF-AMS measurement data of monodisperse SOA particles which represents the initial particle population used for isothermal evaporation measurements.

5. **L93**: Please provide a reference justifying the assumed SOA particle density of 1.5 g cm$^{-3}$.

    Response: We have clarified the text and added the reference.

    Change: Section 2.1 […] Assuming a particle density of 1.5 g cm$^{-3}$ which was estimated on the basis of the elemental ratios of molecular compositions in aerosol particles (Kuwata et al., 2012), …

6. **L94**, **L97**: Is "ug/cm3" a typo (?) (0.1 ug/cm3 = 10,000 ug/m3 )

    Response: Yes, it is typo. We corrected the unit accordingly.

7. **L104**: "gas vapors" seems superfluous

    Response: Thanks for noting this. We changed "gas vapors" to gas phase compounds" .

8. **L119:** Add space between "as" and "("

    Response: We changed it accordingly.

9. **L162**: Please quantify "similar" in this context.

    Response: It is difficult to "quantify" the degree of similarity that the algorithm uses here. Mathematically, the similarity is derived from the correlation of the ion signals. What is deemed acceptably similar by the algorithm depends on the selected number of factors. Ion signals grouped together into a factor in a 4-factor solution are probably split into multiple factors in a 10-factor solution. But still the ion signals in the 4-factor case are more similar to each other than to the other signals in the PMF algorithm.

    However, we added "similar thermal desorption behavior and/or aqueous phase process" in the sentence to provide more context for the term "similar".

10. **L183**: In the OFR254-iN$_2$O mode, NO is generated via N$_2$O + O($^1$D) reactions, not N$_2$O photolysis – see comment #2

    Response: See our response to Comment #2. We changed it accordingly.

11. **L251-L270**: This text should be moved to methods or the supplement

    Response: We decided to move the text "*According to the partitioning theory (Pankow, 1994) for a system … since these particle samples experienced the minimum amount of isothermal evaporation*" into the SI.

12. **L286**: I disagree that a "negligible amount of HNO$_3$ [is] produced in the OFR" – while gas-phase I-CIMS spectra were not presented here, my guess is that IHNO$_3^-$ (or NO$_3^-$) is probably the largest signal in spectra obtained under OFR254-iN$_2$O conditions as it is continuously generated via OH + NO$_2$ . I suggest using the photochemical model to constrain [HNO$_3$] that is obtained at the OFR254-iN$_2$O conditions that were used here, then compare to the HNO$_3$ concentrations that are necessary to initiate/catalyze heterogenous reactions before concluding they are too slow to occur.

    Response: Thank you for the comment. The IHNO$_3^-$ ion was indeed the strongest ion in the gas-phase CIMS spectra. It was so strong that the primary ion (I$^-$) was depleted by orders of magnitude when sampling gas phase directly from the OFR outlet. As this will strongly impact the ionization mechanism, we did not dare to interpret the gas-phase spectra for these experiments. We removed our previous statement "*negligible amount of HNO$_3$*

*[is] produced in the OFR*" and followed the reviewer's suggestion to use the KimSim model to estimate the $HNO_3$ concentration produced in the OFR.

The model-estimated $HNO_3$ concentration is 927 ppb for the high-$NO_x$ cases. Based on the coefficient for partitioning of gaseous $HNO_3$ to SOA described by Ranney and Ziemann (2016), the estimated particulate nitrate (i.e., $[HNO_3]_{SOA} + [NO_3^-]_{SOA}$) concentration is 0.20 μg m$^{-3}$ for the polydisperse SOA (139 ± 29 μg m$^{-3}$) that was formed under the high-$NO_x$ condition in the OFR. With similar gaseous concertation of $HNO_3$ (~ 1 ppm), Ranney and Ziemann (2016) observed 7.5 μg m$^{-3}$ particulate nitrate in the presence of 3000 μg m$^{-3}$ dry SOA that was formed from n-pentadecane with OH radicals. They also found that under such condition, 40% of the cyclic hemiacetals in the SOA can undergo particle-phase dehydration within three hours, with a dehydration rate constant ($k_d$) of 0.25 h$^{-1}$. Note that the α-pinene SOA in our study are highly oxidized and thus exhibit much higher polarity than the n-pentadecane SOA. It is likely that the $k_d$ for the hemiacetals in the α-pinene SOA in our study is several orders magnitude smaller than the value reported for the cyclic hemiacetals in the SOA (Ranney and Ziemann, 2016).

We changed the section in the manuscript to reflect the KinSim values for $HNO_3$.

Change: Section 3.3 […] Due to the lack of direct gaseous $HNO_3$ measurement, we used the KinSim model to estimate the concentration of gaseous $HNO_3$ produced under high-$NO_x$ condition in the OFR. The gaseous $HNO_3$ concertation was estimated to be 927 ppb. Based on the coefficient for partitioning of gaseous $HNO_3$ to SOA described by Ranney and Ziemann (2016), the estimated particulate nitrate (i.e., $[HNO_3]_{SOA} + [NO_3^-]_{SOA}$) concentration is 0.20 μg m$^{-3}$ for the polydisperse SOA (139 ± 29 μg m$^{-3}$) that was formed under the high-$NO_x$ condition in the OFR. With similar gaseous concertation of $HNO_3$ (~ 1 ppm), Ranney and Ziemann (2016) observed 7.5 μg m$^{-3}$ particulate nitrate in the presence of 3000 μg m$^{-3}$ dry SOA that was formed from n-pentadecane with OH radicals. They also found that under such condition, 40% of the cyclic hemiacetals in the SOA can undergo particle-phase dehydration within three hours, with a dehydration rate constant ($k_d$) of 0.25 h$^{-1}$. Note that the α-pinene SOA in our study are highly oxidized and thus exhibit much higher polarity than the n-pentadecane SOA. It is likely that the $k_d$ for the hemiacetals in the α-pinene SOA in our study is several orders magnitude smaller than the value reported for the cyclic hemiacetals in the SOA (Ranney and Ziemann, 2016). […]

13. **L446-L451**: It would be useful to add a brief discussion of the atmospheric implications of these results, especially in regards to the higher evaporation rate of $C_xH_yO_zN$ compounds (relative to $C_xH_yO_z$) that are formed from BVOC oxidation in the presence of NOx and what this means for the SOA formation potential in (sub)urban regions compared to pristine conditions.

Response: We have added a following discussion into the Conclusions.

Change:

4. Conclusions

[…]

In the high-$NO_x$ case, up to 20 wt % of the particle-phase material could be attributed to ON. In general, ON showed approximately 5 °C lower $T_{50}$ and slightly higher volatility, compared with CHO compounds in the high-

NO$_x$ samples. Although the signal contribution of ON differed between the SOA particles that were formed under low- and high-NO$_x$ conditions in the OFR, […]

[…] In addition, particulate ON is highly prevalent in suburban areas, contributing on average 21% of non-refractory submicron particulate matter in mass (Kiendler-Scharr et al., 2016). ON can have sufficiently low volatility to remain in the particle phase. On the other hand, ON is effectivity water-labile and very likely undergoes hydrolysis at the atmospheric-relevant RH (> 40%). Therefore, ON which has low volatility can still be removed from the particle phase, when being hydrolyzed into smaller products that have higher volatility. […]

14. **Figure 1** -"Volume Fraction Remaining" axis label is ambiguous - I know from reading the paper that this refers to the SOA VFR, but someone who just looks at the figure might not make this connection. "RTC" is not defined in the caption text, and it may not be obvious to the reader that "Residence Time" refers to the RTC residence time - please clarify this. The legend and/or caption needs to be clarified to indicate that "low-NOx" and "high-NO$_x$" labels refer to the photochemical conditions in the OFR, rather than RTC conditions; and "RH" should be added to the "dry (<7%)" label

Response: We clarified the caption of Figure 1 following the reviewer's suggestions.

Change:

[Figure]

**Figure 2.** Volume Fraction Remaining (VFR) derived from particle size measurements as function of residence time (t$_R$) in the RTC for α-pinene SOA particles which were formed under low-NO$_x$ (grey) and high-NO$_x$ (orange) conditions in the OFR and evaporated under low RH (< 7 %, squares), intermediate RH (40 % RH, diamonds) and high RH (80 % RH, circles) conditions. The blue and brown areas indicate the collection periods of FIGAERO-CIMS corresponding to fresh and RTC samples.

15. **Figure 1**: I know what an evapogram is, but as far as I can tell this term is not formally defined in the manuscript.

Response: We realized that the use of the term "evapogram" is no longer necessary after we changed the caption of Figure 1.

16. **Figure 1**: It would be useful to overlay a subset of "evapograms" from Buchholz et al. (2019) and/or Li et al. (2021) (or related studies). Alternatively, a table could be added to compare evaporation rates across these studies and others (e.g. Vaden et al.), perhaps by treating the SOA evaporation a first order process for comparison purposes.

Response: We compared the evapograms (i.e., timeseries of volume fraction remaining during the isothermal evaporation) from our study with those in Buchholz et al. (2019) and Vaden et al. (2011), as shown in the Figure below. In each subfigure, we compared our low- and high-NO$_x$ cases with one case of the other studies. In addition, we also performed biexponential fits in the form of $y = 1 + Aexp(-at) + Bexp(-bt)$ with the experimental data. When analyzing two very close evaporation curves, we tried to the same set of A and B in the fitting so that we can qualitatively compare each other. The fitting parameters are summarized in the Table R2 below. A comparison figure is added to the SI and we now refer to these studies when describing our evapogram results in section 3.1.

[Figure]

**Table R2**. List of fitting parameters used in the biexponential fits for the timeseries of volume fraction remaining during the isothermal evaporation.

| Study | System | O:C | RH | A | a × 10$^5$ | B | b × 10$^6$ |
|---|---|---|---|---|---|---|---|
| Low RH | | | | | | | |
| This Study | α-pinene + OH | 0.74 | < 7% | 0.1 | 7.1 | -0.2 | -4.1 |
| This Study | α-pinene + OH + NO$_x$ | 0.77 | < 7% | 0.1 | 7.7 | -0.2 | -1.4 |
| Buchholz et al. (2019) | α-pinene + OH | 0.53 | < 2% | 0.3 | 3.2 | -0.6 | 3.0 |
| Buchholz et al. (2019) | α-pinene + OH | 0.69 | < 2% | 0.1 | 22.5 | -0.2 | -11.1 |
| Buchholz et al. (2019) | α-pinene + OH | 0.96 | < 2% | 0.1 | 31.8 | -0.1 | 2.2 |

| | | | | | | | |
|---|---|---|---|---|---|---|---|
| Vaden et al. (2011) | α-pinene + O$_3$ + cyclohexane (OH scavenger) | N/A | Low RH | 0.3 | 6.3 | -0.6 | -1.5 |
| Intermediate RH | | | | | | | |
| This Study | α-pinene + OH | 0.74 | 40% | 0.3 | 3.7 | -0.6 | 0.2 |
| This Study | α-pinene + OH + NO$_x$ | 0.77 | 40% | 0.3 | 7.1 | -0.6 | 4.9 |
| Buchholz et al. (2019) | α-pinene + OH | 0.53 | 40% | 0.3 | 57.2 | -0.6 | 1.0 |
| Buchholz et al. (2019) | α-pinene + OH | 0.69 | 40% | 0.3 | 23.3 | -0.6 | 5.2 |
| Buchholz et al. (2019) | α-pinene + OH | 0.96 | 40% | 0.3 | 3.6 | -0.6 | 3.8 |
| High RH | | | | | | | |
| This Study | α-pinene + OH | 0.74 | 80% | 0.3 | 4.5 | -0.6 | 0.4 |
| This Study | α-pinene + OH + NO$_x$ | 0.77 | 80% | 0.3 | 4.5 | -0.6 | 0.4 |
| Buchholz et al. (2019) | α-pinene + OH | 0.53 | 80% | 0.3 | 17.5 | -0.6 | -0.3 |
| Buchholz et al. (2019) | α-pinene + OH | 0.69 | 80% | 0.3 | 8.9 | -0.6 | 1.1 |
| Buchholz et al. (2019) | α-pinene + OH | 0.96 | 80% | 0.2 | 0.6 | -0.4 | -16.2 |

Change: Section 3.1

[…] For comparing VFR values between studies, it is important to take particle composition into account (i.e., precursor compounds and oxidation level). The observed evaporation behavior fell between the medium- and high-O:C cases reported in Buchholz et al., 2019 (see Figure S1 in the Supplement), as expected from the measured O:C levels in this study.

SI

[Figure]

Figure S1. Comparison of evaporation behavior of α-pinene SOA particles in different studies, shown as VFR as function of evaporation time.

17. **Figure 2e**: Clarify that the x-axis label refers to **SOA** chemical composition.

Response: We labeled the x-axis in Fig 2e with "SOA Particle Chemical Composition".

Change: See the response to the comment #19.

18. **Figure 2 caption, line 1**: Indicate that thermograms shown here and elsewhere were obtained from the FIGAERO-CIMS.

Response: We clarified the description in the caption.

Change: See the response to the comment #19.

19. **Figure 2 caption, line 2**: add "RH" or perhaps change to "low RH" to more closely parallel "high RH (80% RH)" conditions. or change "high RH" to "humid"

Response: We decided to change the "dry" to "low RH" in Figure 2 and other figures.

Change:

[Figure]

**Figure 3.** FIGAERO-CIMS sum thermograms (STG) (a – d) plotted against desorption temperature ($T_{desorp}$) and the corresponding median desorption temperature ($T_{50}$, diamonds) (e) for the high-NO$_x$ case under low RH (RH < 7 %) and high RH (80 % RH) conditions. CHO compounds and ON are indicated by $C_xH_yO_z$ and $C_xH_yN_{1,2}O_z$, respectively. On the panels (a – d), the solid black lines indicate the total signals of STGs with the green and yellow areas marking the contributions of $C_xH_yO_z$ and $C_xH_yN_{1,2}O_z$ to the STGs, respectively., The gray-striped areas represent the differences in STGs between fresh and RTC stages. The color bands on the abscissa indicate volatility classes. Note that we presented the STGs of RTC stages after accounting for changes in the average VFR (VFR$_{avg}$) between fresh and RTC stages during the FIGAERO sample time.

20. **Figure 3**: Here again, and elsewhere in the text and other figures, I would note that the "low-NO$_x$" and 'high-NO$_x$" labels refer to the OFR conditions rather than the RTC conditions, and note that the "desorption temperature" is associated with the FIGAERO-CIMS.

   Response: We clarified the description in the caption.

   Change: **Figure 4.** Factor thermograms of the five sample factors from the 12-factor PMF solution of CHO compounds in α-pinene SOA particles that were formed under low-NO$_x$ (a) and high-NO$_x$ (b) conditions in the OFR. In addition, thermograms of the sum of ON (sum of NF1 – 3) are shown as purple areas in panel (b). In both panels, the ranges of different volatility classes are highlighted as color bands on the abscissa.

21. **Figures 3 and 5**: With Fig 1 already in place to show how volatility information is extracted from thermograms, I think adding more thermograms in the main paper makes Figures 3 and 5 unnecessarily complex. I would instead show something more like Fig 1e here, i.e., a 4-panel figure plotting T$_{50}$ for each of the factors, one panel each for low-NO$_x$/fresh, low-NO$_x$/RTC, high-NO$_x$ fresh, high-NO$_x$ Then add the factor thermograms to the supplement for the advanced reader.

   Response: We disagree with moving Figures 3 and 5 to the supplement. The information presented Figures 3 and 5 is not the same as that in Figure 1. The Figure 1 shows the volatility information derived from the SMPS measurement, which represent the particle bulk volatility. However, Figures 3 and 5 provides the thermal

desorption behavior of samples factors on the FIGAERO-CIMS as well as factor contributions to the samples under two different formation conditions in the OFR or four different evaporation conditions. Compared with the way recommend by the reviewer, how data is being currently presented in Figures 3 and 5 provide better readability. Furthermore, the information about the $T_{50}$ for two SOA types under four different evaporation conditions is shown in the Figure 7(a) and 8(a). Additionally presenting the shapes of the factor thermograms helps the reader follow the discussion about identifying factors affected by aqueous phase process.

22. **Figures 4 and 5**: What does "normalized fraction" mean, and why is it negative for some species? I assume these figures are showing difference spectra that subtract "fresh" spectra from "RTC" spectra (?) but this should be clarified.

Response: We apologize for the misunderstanding. Now we use "Fraction of Signal" as the y axis for clarity. The negative portion of each mass spectra shows the relative intensity of $C_{10}H_yO_z$ in Figure 4 and $C_{10}H_yNO_z$ in Figure 5 but not the difference spectra between two conditions. This graphic representation was chosen to enhance the separation of the different ion groups. If plotted in the same direction, $C_{10}H_yO_z$ and $C_{6-9}H_yO_z$ would overlap. The shift towards higher carbon numbers (more $C_{10}H_yO_z$) for F3 would no longer be visible. We improved the caption of the two figures to prevent such misunderstandings.

Change:

[Figure]

**Figure 5**. Normalized factor mass spectra of the five sample factors from the 12-factor PMF solution of CHO organics in α-pinene SOA particles that were formed under two $NO_x$ conditions in the OFR. For readability, $C_{10}H_yO_z$ ions are shown as negative values. For each factor mass spectrum, its signal-weighted molecular composition, molecular weight (MW), and oxidation state ($OS_c$) are shown on the right. Right next to each factor label, the squares are shown in the same color scheme as the factor thermograms in Figure 4 to indicate different sample factors. The blue dashed line indicates the average MW of each factor.

[Figure]

**Figure 6.** Factor thermograms (a) and normalized mass spectra (b) for the three sample factors from the eight-factor solution of ON in the α-pinene SOA particles that were formed under high-$NO_x$ conditions in the OFR. In panel (a), the ranges of different volatility classes are indicated in color bands on the abscissa. In panel (b), the bottom portion of each mass spectra represents $C_{10}H_yNO_z$. The signal-weighted molecular composition, molecular weight (MW), and oxidation state ($OS_c$) for each factor are shown on the right. Right next to each factor label in panel (b), the squares are shown in the same color scheme as the factor thermograms in panel (a) to identify different sample factors. The blue dashed line indicates the average MW of each factor.

23. **Figure 4**: It seems more accurate to refer to the "average m/z" rather than "average MW" because a) the FIGAERO-CIMS is not necessarily detecting all of the SOA mass and b) thermal decomposition of larger-MW products may bias low the calculated MW.

Response: We understand the reviewer's concerns. But we would still prefer using "average MW". In the manuscript, we have clearly stated that this average MW was derived on the basis of ions detected with the FIGAERO-CIMS and then used for PMF factors. We do not make any statements about the average MW of total SOA mass. We do not use m/z anywhere else in the manuscript and the quantity molecular weight feels more intuitive for the general reader. We have now clarified / underlines this in the text (see new Figure caption in response to comment #22).

24. **Figure 4:** I suggest adding the "$T_{50}$" (and/or C*) value for each factor to the legend. Along with earlier Figure 3 comments, this addition to Figure 4 might allow the authors to move Figure 3 entirely to the supplement.

Response**:** We disagreed with the reviewer. Just adding the $T_{50}$ and/or C* in Figure 4 does not provide the same level of details as the Figure 3. For example, F2 and F3 have very closed $T_{50}$ values (Table 2). However, their NCR evolution patterns with increasing evaporation and/or RH differed from each other. The factor thermograms in Figure 3 provide the information about the shape and relation to each other which is important for understanding how they are affected by aqueous phase processes for the two SOA types.

25. **Figure 6**: This figure would be easier to interpret if it the signal fractions for F1-F5 in low-NO$_x$ and high-NO$_x$ OFR cases were presented as pie charts. Each pie chart could then just show the total "estimated mass concentration" for the low- and high-NO$_x$ SOA above or below it.

    Response: To compare the signal factions for F1-F5 in two NO$_x$ cases, using the pie chart will not be as straightforward as the slider chart that is currently used in Figure 6a. It is much clearer to see the fractional difference of a factor between two NO$_x$ cases by looking at two circles in a row in the slider chart, compared with looking at two slices in two pie charts if multiple of the slices are changing at the same time. We also prefer the current depiction since it shares some similarities with volatility distributions. The apparent volatility of the factors increases from F1 to F5.

26. **L251-L270 and Figure 6**: It's not clear to me why the summed gas + condensed phase signal is derived in the text and referred to in the figure when the separate gas/particle phase partitioning fractions are never discussed? Presenting and discussing the fraction of signal in gas and condensed phases as a function of C$_{OA}$ seems like it would be a logical extension of the volatility information obtained from the FIGAERO thermograms.

    Response: The use of the summed gas + condensed phase is to understand the impact of NO$_x$ on the amounts of sample factors that were produced in the OFR. Due to the lack of gas-phase CIMS measurements, we are not able to get the fraction of signal in the gas phase. In such case, only discussing the signal fraction of condensed phase is not appropriate. Assume that when the production of a factor decreases due to the perturbation of NO$_x$ in the OFR, the fractions of all remaining factors will increase. This will bias the interpretation with regard to the role of NO$_x$ on the production of compounds in different sample factors during the SOA formation in the OFR.

    In order to understand the effect of effect of NO$_x$ on the α-pinene SOA formation, we decide to first calculate the absolute signal of sample factors in the condensed phase, with the use of the low-RH, fresh samples that were subject to the least change during evaporation. Then we back-calculated the corresponding gas phase concertation for each factor, under the assumption of instantaneous gas-particle partitioning. Note that with regard to the volatility range that sample factors displayed, nearly 100% of compounds in each factor stayed in the particle phase under the experimental mass loadings of (monodisperse) SOA particles. By comparing the calculated summed gas + condensed phase signal for a factor between two NO$_x$ cases, we can know the role of NO$_x$ on the production of this factor. The analysis was possible as we used comparable concentrations of α-pinene for SOA production in the OFR under the two NO$_x$ conditions.

27. **Figure 7:** Why are there only 3 symbols for F1 (no 'high RH RTC case') but 4 symbols for each of the other factors?

    Response: We forgot to add the description in regard to the symbols for F1.

    Change: Figure 7 […] For F1, its factor thermogram contributes less than 5% to the total sample signal and does not show a clear maximum under the high-RH, RTC condition. Therefore, its characteristic T$_{desorp}$ value was not estimated in panel (a) and its NCR was be indicated by a cross close to the 0.1 in panel (b).

28. **Figure 7a**: Why not put SVOC, LVOC, ELVOC text labels at the top of the figure along with the colored bars (as was done with Fig 2)? Similarly, it might be useful to show C* on a secondary axis parallel to T$_{50}$ as was done earlier.

Response: We agreed with the reviewer and accordingly modified Figures 7 that now combines the previous Figures 7 and 8.

Change:

[Figure]

**Figure 7**. Volatility and changes in factor contribution for the five CHO and three ON sample factors and total ON (sum of ON sample factors). Panel (a): Characteristic desorption temperature (characteristic $T_{desorp}$). The marker indicates the $T_{50}$ values, and the horizontal lines mark the interquartile range of the factor thermograms. Panel (b): Net change ratio (NCR) with error bars indicating the uncertainty stemming from the estimated range in molecular weight and particle density. Panel (c): average volume fraction remaining during sample collection (VFR$_{avg}$) with error bars indicating the minimum and maximum values. In all panels, the values for the high-NO$_x$ case are shown with colored markers while the low-NO$_x$ ones are displayed in grey. The colors indicate the sample type. The order of samples is identical for the low-NO$_x$ and high-NO$_x$ data. In panel (a), the range of volatility classes are highlighted with background colors. In panel (b), the dashed line at NCR equal to 1 indicates that any loss is counterbalanced by production, or no change occurs. The two dotted lines at NCR equal to 0.5 and 2 represent significant net loss and production. For F1, its factor thermogram contributes less than 5% to the total sample signal and does not show a clear maximum under the high-RH, RTC condition. Therefore, its

characteristic $T_{desorp}$ value was not estimated in panel (a) and its NCR was be indicated by a cross close to the 0.1 in panel (b).

29. **Figure 7b**: What do the 'x' symbols represent in the top (F1) panel?

Response: We forgot to add the description in regard to the symbols for F1.

Change: **Figure 7** […] For F1, its factor thermogram contributes less than 5% to the total sample signal and does not show a clear maximum under the high-RH, RTC condition. Therefore, its characteristic $T_{desorp}$ value was not estimated in panel (a) and its NCR was be indicated by a cross close to the 0.1 in panel (b).

30. Can you come up with a brief name/description for each of the factors so that when information about F1-F5, etc. are presented in subsequent figures, it's easier to make a connection as to what they represent?

Response: Unfortunately, it is very hard to logically name each factor. The role of a factor during particle evaporation varied with $NO_x$ conditions in the OFR or with RH and/or time during evaporation. It means that a factor can be an "educt" factor under one condition but also can be a "product" factor under another condition. The current factor labels are based on their apparent volatility as seen in the fresh, low-RH sample which had the least extent of evaporation.

31. Repeatedly using the terms "non-nitrated" and "nitrated" is cumbersome – perhaps consider using "$C_xH_yO_z$" and "$C_xH_yO_zN$" descriptors instead (after defining them once).

Response: It is indeed cumbersome. To make these two terms more distinguishable from each other, we decided to use CHO compounds and ON to indicate non-nitrated organics and organic nitrates in the corrected manuscript, respectively. We adjusted the main text, figures, and the SI with these new labels wherever appropriate.

**Reference**

Buchholz, A., Lambe, A. T., Ylisirniö, A., Li, Z., Tikkanen, O.-P., Faiola, C., Kari, E., Hao, L., Luoma, O., Huang, W., Mohr, C., Worsnop, D. R., Nizkorodov, S. A., Yli-Juuti, T., Schobesberger, S., and Virtanen, A.: Insights into the O : C-dependent mechanisms controlling the evaporation of α-pinene secondary organic aerosol particles, Atmos. Chem. Phys., 19, 4061-4073, 10.5194/acp-19-4061-2019, 2019.

Kiendler-Scharr, A., Mensah, A. A., Friese, E., Topping, D., Nemitz, E., Prevot, A. S. H., Aijala, M., Allan, J., Canonaco, F., Canagaratna, M., Carbone, S., Crippa, M., Dall Osto, M., Day, D. A., De Carlo, P., Di Marco, C. F., Elbern, H., Eriksson, A., Freney, E., Hao, L., Herrmann, H., Hildebrandt, L., Hillamo, R., Jimenez, J. L., Laaksonen, A., McFiggans, G., Mohr, C., O'Dowd, C., Otjes, R., Ovadnevaite, J., Pandis, S. N., Poulain, L., Schlag, P., Sellegri, K., Swietlicki, E., Tiitta, P., Vermeulen, A., Wahner, A., Worsnop, D., and Wu, H. C.: Ubiquity of organic nitrates from nighttime chemistry in the European submicron aerosol, Geophys. Res. Lett., 43, 7735-7744, 10.1002/2016gl069239, 2016.

Kuwata, M., Zorn, S. R., and Martin, S. T.: Using elemental ratios to predict the density of organic material composed of carbon, hydrogen, and oxygen, Environ. Sci. Technol., 46, 787-794, 10.1021/es202525q, 2012.

Lambe, A., Massoli, P., Zhang, X., Canagaratna, M., Nowak, J., Daube, C., Yan, C., Nie, W., Onasch, T., and Jayne, J.: Controlled nitric oxide production via O($^1$D)+ N$_2$O reactions for use in oxidation flow reactor studies, Atmos. Meas. Tech., 10, 2283-2298, 2017.

Li, Z., Buchholz, A., Ylisirniö, A., Barreira, L., Hao, L., Schobesberger, S., Yli-Juuti, T., and Virtanen, A.: Evolution of volatility and composition in sesquiterpene-mixed and α-pinene secondary organic aerosol particles during isothermal evaporation, Atmos. Chem. Phys., 21, 18283-18302, 10.5194/acp-21-18283-2021, 2021.

Palm, B. B., Campuzano-Jost, P., Day, D. A., Ortega, A. M., Fry, J. L., Brown, S. S., Zarzana, K. J., Dube, W., Wagner, N. L., and Draper, D. C.: Secondary organic aerosol formation from in situ OH, o 3, and no 3 oxidation of ambient forest air in an oxidation flow reactor, Atmos. Chem. Phys., 17, 5331-5354, 2017.

Pandis, S. N. and Seinfeld, J. H.: Atmospheric chemistry and physics: From air pollution to climate change, Wiley, 2006.

Peng, Z. and Jimenez, J. L.: Kinsim: A research-grade, user-friendly, visual kinetics simulator for chemical-kinetics and environmental-chemistry teaching,  2019.

Peng, Z., Lee-Taylor, J., Orlando, J. J., Tyndall, G. S., and Jimenez, J. L.: Organic peroxy radical chemistry in oxidation flow reactors and environmental chambers and their atmospheric relevance, Atmos. Chem. Phys., 19, 813-834, 2019.

Ranney, A. P. and Ziemann, P. J.: Kinetics of acid-catalyzed dehydration of cyclic hemiacetals in organic aerosol particles in equilibrium with nitric acid vapor, J. Phys. Chem. A, 120, 2561-2568, 10.1021/acs.jpca.6b01402, 2016.

Vaden, T. D., Imre, D., Beranek, J., Shrivastava, M., and Zelenyuk, A.: Evaporation kinetics and phase of laboratory and ambient secondary organic aerosol, Proc. Natl. Acad. Sci. U. S. A., 108, 2190-2195, 10.1073/pnas.1013391108, 2011.

Wang, Y., Piletic, I. R., Takeuchi, M., Xu, T., France, S., and Ng, N. L.: Synthesis and hydrolysis of atmospherically relevant monoterpene-derived organic nitrates, Environ. Sci. Technol., 55, 14595-14606, 10.1021/acs.est.1c05310, 2021.

---

## Author Response (AR2)

**Reply To Reviewer 1**

We want to thank the reviewer for accepting our manuscript

**Reply To Reviewer 2**

We would like to thank Dr. Lambe for his insightful comments. Below we give point-by-point responses to the comments.

1. L93 – If the OFR254 mode is being used, I don't understand how the calculated OHexp value is now higher in the high-NOx condition relative to the low-NOx condition (in the absence of other changes): N2O creates an additional O(1D) sink, and NO2 creates additional OH reactivity. In the discussions paper, the OHexp values were 2.6e11 (low-NOx) and 1.72e11 (high-NOx); now, they are 1.82e12 (low-NOx) and 2.45e12 (high-NOx). In principle, it is possible to increase the OHexp in OFR185-iN2O due to N2O + hv185 -> O(1D) + N2, but I am not sure how that is possible with OFR254-iN2O. Unless I'm missing something, this doesn't seem right to me. I suggest double checking the model calculations and/or emailing me the KinSim mechanism (if Prof. Ng feels this is appropriate, given that the review is not anonymous) and we can try to straighten this out together.

Response: We double checked the model calculations. Indeed, the OH exposures that were estimated with KimSim were  $1.82 \times 10^{12}$  (low-NOx) and  $2.45 \times 10^{12}$  (high-NOx) molec cm-3 s. We acknowledge that the addition of N2O would decrease the OH production in the OFR without changing the setting of the UV254 lamps. In order to keep the OH exposure close enough between two NOx cases, we intentionally applied higher settings for the OH lamp in the high-NOx case compared to the low-NOx cases. The average photon fluxes were  $1.08 \times 10^{15}$  and  $2.74 \times 10^{15}$  photons cm-2 s-1 for the low- and high-NOx cases, respectively. We added a sentence in section 2.1 to avoid any misunderstanding in the future. The reason for "*the OHexp values were*  $2.6 \times 10^{11}$  (*low-NOx*) and  $1.72 \times 10^{11}$  (*high-NOx*); now, they are 1.82e12 (*low-NOx*) and 2.45e12 (*high-NOx*)." can be due to the differences in the model settings between the simplified OFR model and the KinSim model.

**Changes:**

**Section 2.1**

**[...] For the two NOx cases, the voltage of the 254-nm lamps was adjusted to ensure similar OH exposure. [...]**

2. In regards to the authors' reply to my Comment #1, I will note that the model infrastructure outlined in Section 2.3 of Tikkanen et al. (2020) already exists, and the Kuopio group has the expertise to apply it here. It seems that the authors agree with the essence of my suggestion, but are mostly concerned about implementing it here because of the potential scope of the additional analysis that would be required. I will attempt to try to restrict the scope of what I am asking for in an attempt to make it more tractable, while also providing what I feel will be significant value added to the revised manuscript.

a. Because Tikkanen et al. (2020) already derived volatility distributions from evapograms of alphapinene SOA generated under low-NOx conditions, it would be satisfactory to restrict the analysis to volatility distributions obtained from evapograms of the alpha-pinene SOA generated under high-NOx conditions here and compare to volatility distributions obtained from the corresponding FIGAERO-CIMS thermograms of high-NOx alpha-pinene SOA

Response: We understood the strong wish from Dr. Lambe to compare the volatility distributions (VDs) between evapograms and FIGERO-CIMS thermograms in the same way as Tikkanen et al. (2020). As our manuscript and more detailed discussion below shows, our data and results point out that the aqueous-phase chemistry plays a significant role in this study. Unfortunately, the inverse model in Tikkanen et al., does not include aqueous-phase chemical processes, and as we show below, is not a suitable model for our system. Considering this, and the significant human resources needed to use the optimization algorithm correctly, we suggest an alternative approach.

To provide a comparison of the FIGAERO-CIMS data with the observed isothermal evaporation, we performed forward modelling for particle evaporation at high RH but without including any aqueous-phase chemistry. This can be done by using the VDs from the FIGAERO-CIMS thermograms collected at the fresh and high-RH stage as inputs for the same liquid-like evaporation model (LLEVAP) that was used in Tikkanen et al. (2020). With this setup, particle volatility is presumed as the only driver for particle size change in the simulation. The purpose is to see whether the use of FIGAERO-CIMS-derived VD can reproduce the observed isothermal evaporation at high RH. This approach is the same as that outlined in the last two paragraphs in Section 3.2 and shown in Figure 4 in Tikkanen et al. (2020).

The results of the model simulations are shown in Figure R1 a and b for the high- and low-NOx cases at high RH, respectively. We found that the model simulations underestimated the particle evaporation rate (solid blue line in Fig R1a and b).

In Tikkanen et al. (2020), using the  $T_{50}$  values to characterize the volatility of each factor also led to an underestimation of the extent of evaporation (solid blue line in Figure R1c). But by allowing the desorption temperature of a PMF factor to vary in the range of the 25th and 75th percentile (Section 3.3 and Figure 5 in Tikkanen et al. (2020)), they were able to reproduce the observed evaporation of  $\alpha$ -pinene SOA particles. Their result with the optimized VD is shown as the solid green line below in Fig. R1 c (see also Tikkanen et al. (2020) Figure. 5a). Similar agreement could not be achieved for our data (range given by shaded area in Figure R1a and b). The VD derived from the FIGAERO measurements in the present study is too far in the LVOC or ELVOC range, while 40% of the VD from Tikkanen et al. (2020) can be found in the SVOC range. The uncertainty range estimated from the 25th and 75th percentiles does not increase the volatility enough to match the evaporation observed in the thermograms. Therefore, the discrepancy between the simulated and observed evaporation curves in our study very likely suggests the significant role of aqueous processes for the low- and high-NOx cases. Due to the lack of detailed information about the products resulting from aqueous-phase processes and their volatilities, it is unreasonable to further try to include the aqueous process into the current forward modelling setup. This also means that while the LLEVAP model used in Tikkanen et al. (2020) probably could find a VD solution to reproduce the evaporation curves observed in our study, it would be using incorrect assumptions probably leading to a shift of the modelled VD to higher C\* values.

Figure R1. Measured (circles) and simulated (lines and shaded areas) volume fraction remanning (VFR) as function of resisdence time ( $t_R$ ). The panels (a) and (b) show the data of the high- and low NOx cases, while the panel (c) displays the data of medium O:C case from Tikkanen et al. (2020). In each panel, the simulations were computed with the T50 values (solid blue line) and T25 – T75 ranges (shaded areas in blue) of the volatility distributions obtained from the FIGAERO-CIMS data (VDPMF). The simulation from the optimized VDPMF that was reported by Tikkanen et al. (2020) is shown in solid green line in the panel (c).

 Further, if modeling the aqueous phase chemistry that occurs at high-RH conditions in the RTC is the main issue here, in my opinion, it is sufficient to restrict the analysis to the RH<7% and/or RH=40% cases.

Response: The extent of aqueous-phase chemistry might be smaller at lower RH due to the decreasing amount of particulate water content, compared to that at high RH. However, the effect of particle viscosity becomes more important with decreasing RH. The process in Tikkanen et al. (2020) was to use the high-RH cases to derive the VD and then use that as an input for the KMGAP model which would find the viscosity for the intermediate- and low-RH cases. Therefore, using only low- and intermediate-RH data would not be enough to derive the VD.

- c. In summary, my request would be the following:
  - i. add an abbreviated version of Section 2.3 from Tikkanen et al. (2020) to the methods section of this paper
  - ii. apply the approach of Tikkanen et al. (2020) to the RH<7% and/or RH=40% high-NOx alphapinene SOA evapograms, "fresh" and "RTC" cases that are shown in Fig. 1
  - add one figure analogous to Fig 7. from Tikkanen et al. (2020) that compares volatility distributions obtained from evapograms and thermograms of "fresh" and "RTC" high-NOx SOA, and appropriate text to the discussion.

Response: As stated above, we feel that the model-optimization approach used in Tikkanen et al. (2020) is not required here. The presented analysis using forward modelling is enough to answer our research questions. We added a discussion in Section 3.1 in the main text, and also an abbreviated description of the forward modelling in the section S1.1 in SI.

Changes:

Main Text

Section 3.1

[...]

To investigate if particulate water acted also as a catalyzer for aqueous-phase processes for the lowand high-NOx cases, we calculated the expected isothermal evaporation behavior based on the particlephase volatility distribution from the PMF analysis of the FIGAERO measurements (see section 3.2) by using a liquid-like evaporation model (LLEVAP) (Lehtinen and Kulmala, 2003; Yli-Juuti et al., 2017), which assumes particle volatility as the only driver for particle size change during particle evaporation (see section S1.1 in Supplement). With this method, Tikkanen et al. (2020) could reproduce the observed isothermal evaporation under high-RH conditions for  $\alpha$ -pinene SOA particles if no substantial aqueous-phase processes occurred, and particulate water acted primarily as a plasticizer. But in this study, we found a clear discrepancy between the observed isothermal evaporation and the LLEVAP simulations even when considering the interquartile range of the volatility distribution (see Figure S2 in the Supplement). The volatility distribution derived from the FIGAERO measurement was too far in the LVOC and ELVOC range, i.e., using this volatility distribution always underestimates the amount of isothermal evaporation. This behavior can be interpreted as a sign for aqueous-phase processes. Water acted as a catalyst for reactions creating products of higher volatility which then evaporate from the particle, thus leading to more isothermal evaporation than expected from the original volatility distribution. We discuss more details about the changes in the composition of the residual particles and possible reaction pathways in section 3.4.

**Supporting Information**

**S1.1 Forward modelling for high-RH cases**

To model the particle evaporation at high RH, we applied a liquid-like evaporation model (LLEVAP) (Vesala et al., 1997; Lehtinen and Kulmala, 2003; Yli-Juuti et al., 2017). This model assumes there is no concentration gradient existing in the particle. It solves a series of differential equations which describe mass fluxes of organic compounds between particle and gas phase on the basis of gas-phase diffusion, i.e., the difference in the gas-phase concentration of an organic compound near the particle surface and far away for the particle. In this case, the particle-phase volatility distribution (VD) is the only driving factor for the evaporation rate in the model simulation.

Following Tikkanen et al. (2020), we first derived the VD from the positive matrix factorization (PMF) analysis of the FIGAERO measurements (hereafter VDPMF). Afterwards, the derived VDPMF was used as the initial particle-phase VD in the LLEVAP model. The VDPMF was derived from the median desorption temperature values ( $T_{50}$ ), with the interquartile range of desorption temperature ( $T_{25} - T_{75}$ ) in the factor thermograms as the uncertainty. The start time for each LLEVAP simulation was set to the middle time point (i.e., 15 min) of the FIGAERO measurements. In our LLEVAP simulations, we applied the same set of particle properties as Tikkanen et al. (2020). This includes the gas-phase diffusion coefficient, molar mass, particle density, particle surface tension, and mass accommodation coefficient. For each of the two NOx cases, the LLEVAP simulations for  $\alpha$ -pinene SOA particle (O : C = 0.69) from Tikkanen et al. (2020), shown in Figure S2c (see also Figure 5a in Tikkanen et al. (2020)). Additionally, Tikkanen et al. (2020) found the best reconstruction of the measured particle evaporation by allowing each PMF factor to vary in the range of  $T_{25} - T_{75}$  (VDPMF, optimized, green solid line in Fig S2c).

---

## Author Response (AR3)

**Reply To Reviewer 2**

We would like to again thank Dr. Lambe for his comment. Below we provide our response to the comment

1. L91 - for the reader's reference, please state the lamp voltage values and/or corresponding photon fluxes that were used to offset OH suppression following NOx generation.

Response: We added the following sentences in the main text to provide details about the lamp adjustment and add one more row about the photon flux in the Table S1.

Change:

Section 2.1

[...] We indirectly determined the photon fluxes for each experiment using the measured O3 decay without any  $\alpha$ -pinene and N2O in the OFR. With the KinSim model developed by Peng and Jimenez (2019), we estimated the photon flux by varying the model input photon flux until the model output O3 concentration agreed with the measured one. The estimated photon fluxes were  $(1.08 \pm 0.14) \times 10^{15}$  and  $(2.74 \pm 0.35) \times 10^{15}$  photons cm-2 s-1 for the low- and high-NOx cases, respectively. Furthermore, [...]

Table S1. Summary of experimental conditions and results of  $\alpha$ -pinene SOA generation

|                                                                                          | Low-NO x | High-NO x |
|------------------------------------------------------------------------------------------|---------------------|----------------------|
| [VOC] OFR (ppb) a                                                  | $254 \pm 11$        | 296 ± 14             |
| [N 2 O] OFR (%)                                                    | N/A                 | $1.82\pm0.10$        |
| [O 3 ] OFR (ppm) b                                      | $9.76 \pm 0.31$     | $6.85 \pm 0.36$      |
| photon flux
(10 15 photons cm -2 s -1 ) b | $1.08\pm0.14$       | $2.74 \pm 0.35$      |
| T OFR (°C)                                                                    | $24.66\pm0.76$      | $28.14 \pm 0.91$     |
| RH OFR (%)                                                                    | $44.19 \pm 2.17$    | 38.74 ± 2.63         |
| nominal residence time (s)                                                               | 160                 | 160                  |
| effective OH exposure
(10 12 molec cm -3 s) c        | $1.82 \pm 0.21$     | 2.45 ± 0.09          |
| $\frac{[RO_2] + [NO]}{[RO_2] + [HO_2]}$                                                  | N/A                 | $0.84 \pm 0.19$      |

| fraction Loss to OH (%)             | 96            | 50            |
|-------------------------------------|---------------|---------------|
| fraction Loss to O 3 (%) | 4             | 1             |
| fraction Loss to $NO_3$ (%)         | 0             | 49            |
| oxygen-to-carbon (O:C) d | $0.77\pm0.03$ | $0.74\pm0.01$ |
| oxidation state $(OS_c)^d$          | $0.05\pm0.04$ | $0.02\pm0.02$ |

a Mixing ratio of  $\alpha$ -pinene was corrected with the dilution of O3-contained flow but without the loss due to pure ozonolysis at the inlet. b O3 was measured at the OFR outlet after 254-nm UV lamps were switched on but without the addition of  $\alpha$ -pinene and N2O. The photon flux was estimated by varying the model input photon flux in the KimSim model (Peng and Jimenez, 2019) until the model output O3 concentration agreed with the measured one. c OH exposure was calculated with the KinSim model (Peng and Jimenez, 2019). d The values of the oxygen to carbon ratio (O:C) and the oxidation state (OSc) were derived from the HR-ToF-AMS measurement data of monodisperse SOA particles which represents the initial particle population used for isothermal evaporation measurements.

**References**

Peng, Z. and Jimenez, J. L.: Kinsim: A research-grade, user-friendly, visual kinetics simulator for chemical-kinetics and environmental-chemistry teaching, 2019.